# Multifaceted regulation of hepatic lipid metabolism by YY1

Gang Pan[1] , Klev Diamanti[1,2], Marco Cavalli[1], Ariadna Lara Gutiérrez[1] , Jan Komorowski[2,3,4,5], Claes Wadelius[1]

**Recent studies suggested that dysregulated *YY1* plays a pivotal role in many liver diseases. To obtain a detailed view of genes and pathways regulated by YY1 in the liver, we carried out RNA sequencing in HepG2 cells after YY1 knockdown. A rigid set of 2,081 differentially expressed genes was identified by comparing the YY1-knockdown samples (n = 8) with the control samples (n = 14). YY1 knockdown significantly decreased the expression of several key transcription factors and their coactivators in lipid metabolism. This is illustrated by YY1 regulating PPARA expression through binding to its promoter and enhancer regions. Our study further suggest that down-regulation of the key transcription factors together with YY1 knockdown significantly decreased the cooperation between YY1 and these transcription factors at various regulatory regions, which are important in regulating the expression of genes in hepatic lipid metabolism. This was supported by the finding that the expression of *SCD* and *ELOVL6*, encoding key enzymes in lipogenesis, were regulated by the cooperation between YY1 and PPARA/RXRA complex over their promoters.**

## Introduction

Liver is one of the most important metabolic organs of the human body. Liver diseases arising from various etiologies account for ~2 million deaths per year worldwide, mostly due to complications of cirrhosis, viral hepatitis, and hepatocellular carcinoma (HCC) (Asrani et al, 2019). Over the past few decades, nonalcoholic fatty liver disease (NAFLD) has emerged as the most prominent cause of chronic liver disease worldwide and occurs in about 25% of the world population (Younossi et al, 2019). NAFLD, characterized by excessive lipid deposition in the hepatocytes of the liver parenchyma, can gradually develop into a series of lipid disorders including steatosis, nonalcoholic steatohepatitis (NASH) which is characterized by varying degrees of fibrosis, that may further progress into cirrhosis and HCC (Asrani et al, 2019). The underlying mechanism of NAFLD is believed to be the hepatic manifestation of the metabolic syndrome that is highly prevalent in obese and diabetic subjects (Asrani et al, 2019; Younossi et al, 2019).

Yin Yang 1 (YY1) is a ubiquitous and multifunctional zinc-finger transcription factor (TF) that can activate or repress gene expression, depending on the cellular context (Zhang et al, 2017). A large number of YY1 target genes have been identified that participate in a broad range of biological process, such as embryogenesis, differentiation, replication, and cellular proliferation (Shi et al, 2015; Zhang et al, 2017). *YY1* is overexpressed in many forms of carcinomas including HCC, the expression of which was commonly correlated with advanced malignancy or poor clinical outcomes (Tsang et al, 2016; Zhang et al, 2017; Sarvagalla et al, 2019). In addition to HCC, cumulative data indicate a pivotal role of YY1 in almost all other liver diseases, for example, hepatitis induced by virus infection, liver fibrosis, liver regeneration and both alcoholic liver disease and NAFLD (Zhang et al, 2017). YY1 has recently been implicated in metabolic dysfunction in the liver including glycometabolism reprograming, lipid metabolism, and bile acid metabolism that is involved in various diseases (Zhang et al, 2017). This may partially explain the observed dysregulation of *YY1* in various liver diseases.

The hallmark of NAFLD is triglyceride (TG) accumulation in the cytoplasm of hepatocytes that is caused by the imbalance between lipid acquisition and removal in the liver (Kawano & Cohen, 2013). As the main building blocks of TG, free fatty acids (FAs) play essential roles in the pathogenesis of NAFLD. The liver acquires free FAs from three major sources that are directly dietary intake, lipolysis of TG in adipose tissue and de novo lipogenesis (DNL) in the liver with carbohydrates, especially fructose, as substrate. The fates of FAs in the liver are mitochondrial fatty acids β-oxidation (FAO) and re-esterification to form TG. TG can then be exported from the liver into the blood as very low-density lipoprotein or stored as lipid droplets (Kawano & Cohen, 2013; Friedman et al, 2018). The overload of free FAs in the liver resulting from either increased intake and/or synthesis or decreased utilization through FAO will increase the synthesis of TG in the liver. This in combination with the decreased efficiency of TG removal through very low-density lipoprotein will induce TG accumulation in the liver that further triggers NAFLD

[1]Science for Life Laboratory, Department of Immunology, Genetics and Pathology, Uppsala University, Uppsala, Sweden  [2]Science for Life Laboratory, Department of Cell and Molecular Biology, Uppsala University, Uppsala, Sweden  [3]Swedish Collegium for Advanced Study, Uppsala, Sweden  [4]Institute of Computer Science, Polish Academy of Sciences, Warsaw, Poland  [5]Washington National Primate Research Center, Seattle, WA, USA

Correspondence: claes.wadelius@igp.uu.se; gang.pan@igp.uu.se

progression. Recent studies have shown that dysregulation of *YY1* is involved in the pathogenesis of NAFLD (Lu et al, 2014; Lai et al, 2018; Yuan et al, 2018). The expression of *YY1* was markedly increased in the livers of NAFLD patients, which is significantly associated with the progression of NAFLD at different stages (Lu et al, 2014; Yuan et al, 2018). By utilizing both animal models and cell line models, YY1 was found to regulate hepatic lipid metabolism either directly activating the DNL pathway or inhibiting the FAO pathway that further induced NAFLD initiation and progression (Lu et al, 2014; Yuan et al, 2018; Li et al, 2019). These studies primarily focused on specific pathways regulated by YY1 in NAFLD. To better understand the various genes and pathways regulated by YY1 in liver, we generated HepG2 cells with down-regulated *YY1* expression. RNA sequencing (RNA-seq) of these samples revealed a rigid set of genes and pathways regulated by YY1. The genes up-regulated by YY1 knockdown were tightly associated with perturbed promoter and enhancer interactions mediated by YY1, whereas the genes down-regulated by YY1 knockdown were significantly enriched in various biological processes in lipid metabolism (Weintraub et al, 2017). Our study further proved that YY1 directly or indirectly regulates the expression of several important TFs and their coactivators in lipid metabolism, which provided novel insights into the molecular mechanisms associated with *YY1* overexpression in NAFLD and other liver diseases.

# Results

### RNA-seq revealed extensive changes in gene expression profile induced by YY1 knockdown

To investigate the role of YY1 in regulating hepatic gene expression, we knocked down *YY1* expression in the liver cancer cell line HepG2 employing both lentiviral-mediated shRNA and artificial miRNA (amiRNA). The expression of *YY1* was significantly down-regulated by YY1 knockdown determined by quantitative reverse transcription polymerase chain reaction (RT-qPCR) assay (Fig 1A). The efficiency of our strategy in knocking down *YY1* expression was further validated by Western blot (Fig 1B). We next sought to determine the differentially expressed genes (DEGs) induced by YY1 knockdown through RNA-seq. Three batches of YY1-knockdown samples (*n* = 8) and four batches of samples transduced with the control virus (*n* = 14) cultured under the same condition were subjected to RNA-seq. By targeted sequencing of the 3′ end of polyadenylated RNA, we obtained an average of 7.2 million reads per sample (range from 5.5 to 8.4). An average of 5.7% of the reads (range from 3.0 to 9.1%) were filtered out by quality trimming. The remaining reads which were properly paired were mapped to the reference human genome (UCSC version hg19). We obtained an average mapping rate of 88.5% (range from 85.0 to 90.7%) for paired reads that are uniquely mapped and an average mapping rate of 9.8% (range from 7.4 to 12.9%) for paired reads that are mapped concordantly more than once (Table S1). The uniquely mapped reads were further mapped to known genes with an average mapping rate of 77.9% (range from 76.0 to 79.8%) by featureCounts (Liao et al, 2014).

Principal component analysis showed that the control samples and YY1-knockdown samples were grouped together and well separated based on the first principal component (Fig 1C). We applied a stringent statistic threshold of greater than $\log_2$ fold change of 0.7 and *P*-value of < 0.001 (corresponding to *P*adj < 0.003) to identify a consensus of DEGs. A total of 2,273 DEGs composed of 1,348 up-regulated genes and 925 down-regulated genes were identified based on Ensembl gene annotations (Fig 1D). The biotypes of the identified DEGs are mainly classified as either protein coding or long noncoding RNAs (Fig 1E). The identified DEGs were further annotated with Entrez Gene database which gave rise to 2,081 DEGs including 1,225 up-regulated and 856 down-regulated genes and were used for further analysis (Table S2) (Maglott et al, 2011). To verify the results of our RNA-seq data, 61 genes including 8 up-regulated, 41 down-regulated and 12 unchanged were selected for RT-qPCR analysis in two batches of samples (*n* = 8) independent from the samples used in RNA-seq and showed similar expression patterns compared with their expression changes observed in RNA-seq (Fig S1A and B). Pearson correlation analysis showed a good correlation (r = 0.92) between RNA-seq data and RT-qPCR results (Table S3 and Fig 1F). These results indicate that our RNA-seq data accurately imaged the transcriptional alternations induced by YY1 knockdown.

### Functional enrichment analysis of the DEGs induced by YY1 knockdown

To get further insights into the biological processes and pathways significantly changed by YY1 knockdown, the DEGs were subjected to Gene Ontology (GO) and Kyoto Encyclopedia of Genes and Genomes (KEGG) pathway enrichment analysis (Yu et al, 2012). The up-regulated genes were significantly enriched in GO terms related to cell differentiation and homeostasis, such as epithelial cell differentiation, vasculature development, chemotaxis, and extracellular matrix organization (Fig 2A). The down-regulated genes were significantly enriched in multiple GO terms related to lipid metabolism, such as steroid biosynthesis process, FA metabolic process, lipid biosynthetic process, cholesterol metabolic process, and lipid transport (Fig 2B). The KEGG pathway enrichment analysis revealed that the up-regulated genes were significantly enriched in pathways such as the Hippo signaling pathway and cellular senescence, whereas the down-regulated genes were significantly enriched in pathways tightly related to lipid metabolism, such as FA metabolism, PPAR signaling pathway, and cholesterol metabolism (Fig 2C and D).

To validate the identified enrichment of GO terms and KEGG pathways in the DEGs, we further carried out gene set enrichment analysis (GSEA) with the C2 gene sets from the Molecular Signature Database (MSigDB) as input (Subramanian et al, 2005). Accordingly, the up-regulated genes were significantly enriched with signatures of genes in extracellular matrix organization and genes regulated by SUZ12, a key component of the polycomb repressive complex 2 (PRC2) (Fig 2E) (Xu et al, 2015). The down-regulated genes were significantly enriched with signatures of genes specifically expressed in the liver organ and genes involved in metabolism of steroid hormones, particularly genes involved in cholesterol biosynthesis (Fig 2F).

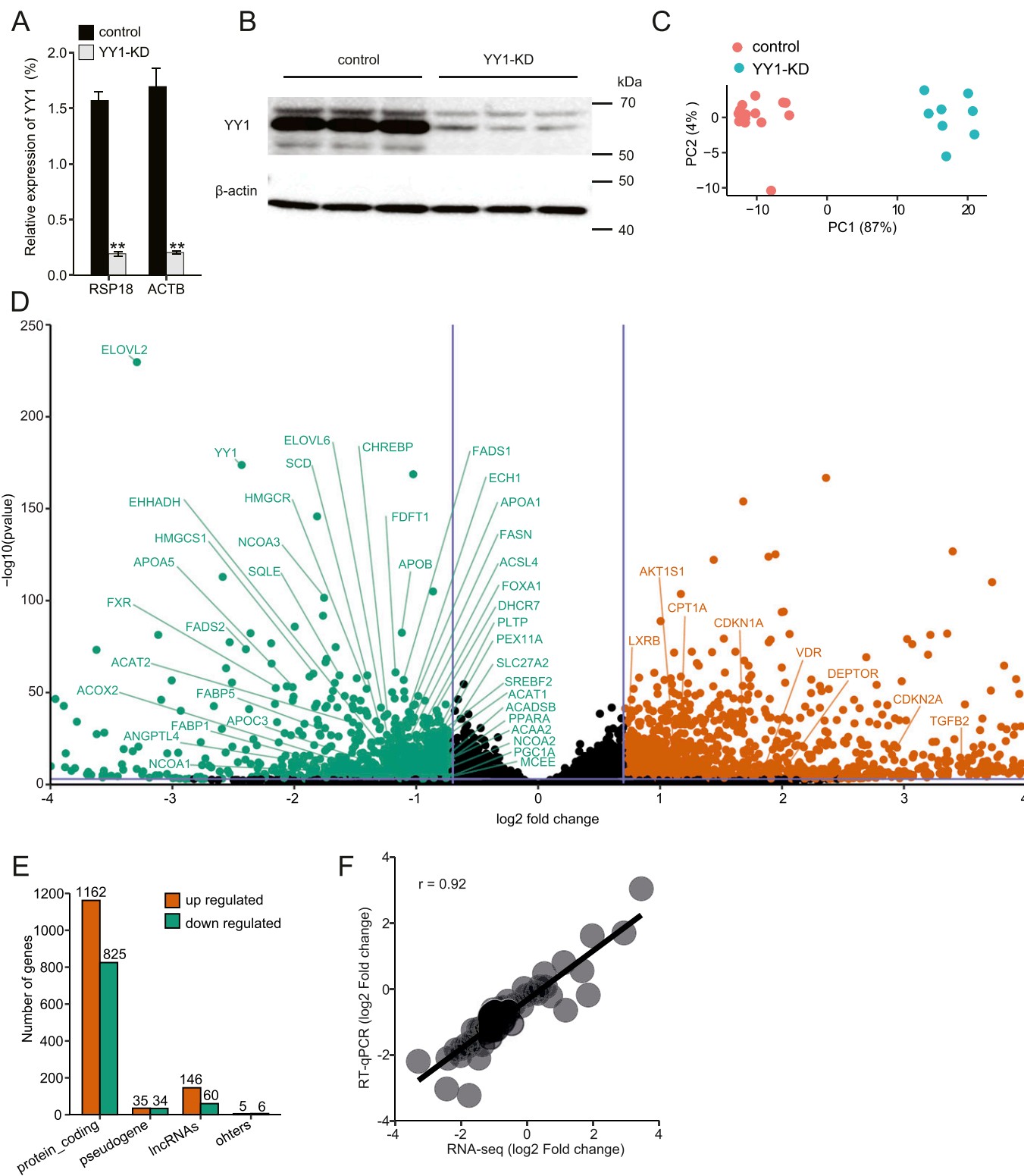

**Figure 1. Global RNA sequencing revealed extensive gene expression changes induced by Yin Yang 1 (YY1) knockdown in HepG2 cells.**
**(A, B)** The efficiency of YY1 knockdown was validated by both RT-qPCR (A) and Western blot (B). The expression of *YY1* in RT-qPCR was normalized with the expression of *RSP18* and *ACTB*, respectively. Error bars, SD; *n* = 8 technical replicates. **P < 0.01 calculated by two-tailed *t* tests. **(C)** Principal component analysis of RNA sequencing (RNA-seq) data. The YY1-knockdown samples (*n* = 8) were clustered together and well separated from the control samples (*n* = 14). **(D)** Volcano plot showing differentially expressed genes (DEGs) induced by YY1 knockdown in HepG2 cells. Threshold of *P*-value < 0.001 and absolute log$_2$ fold change > 0.7 was used to determine the DEGs. The DEGs that are further verified by RT-qPCR are listed. **(E)** Histogram of the biotypes of the identified DEGs based on Ensembl annotation. **(F)** Correlation of gene expression levels between RNA-seq and RT-qPCR data. The detailed expression changes are listed in Table S3.

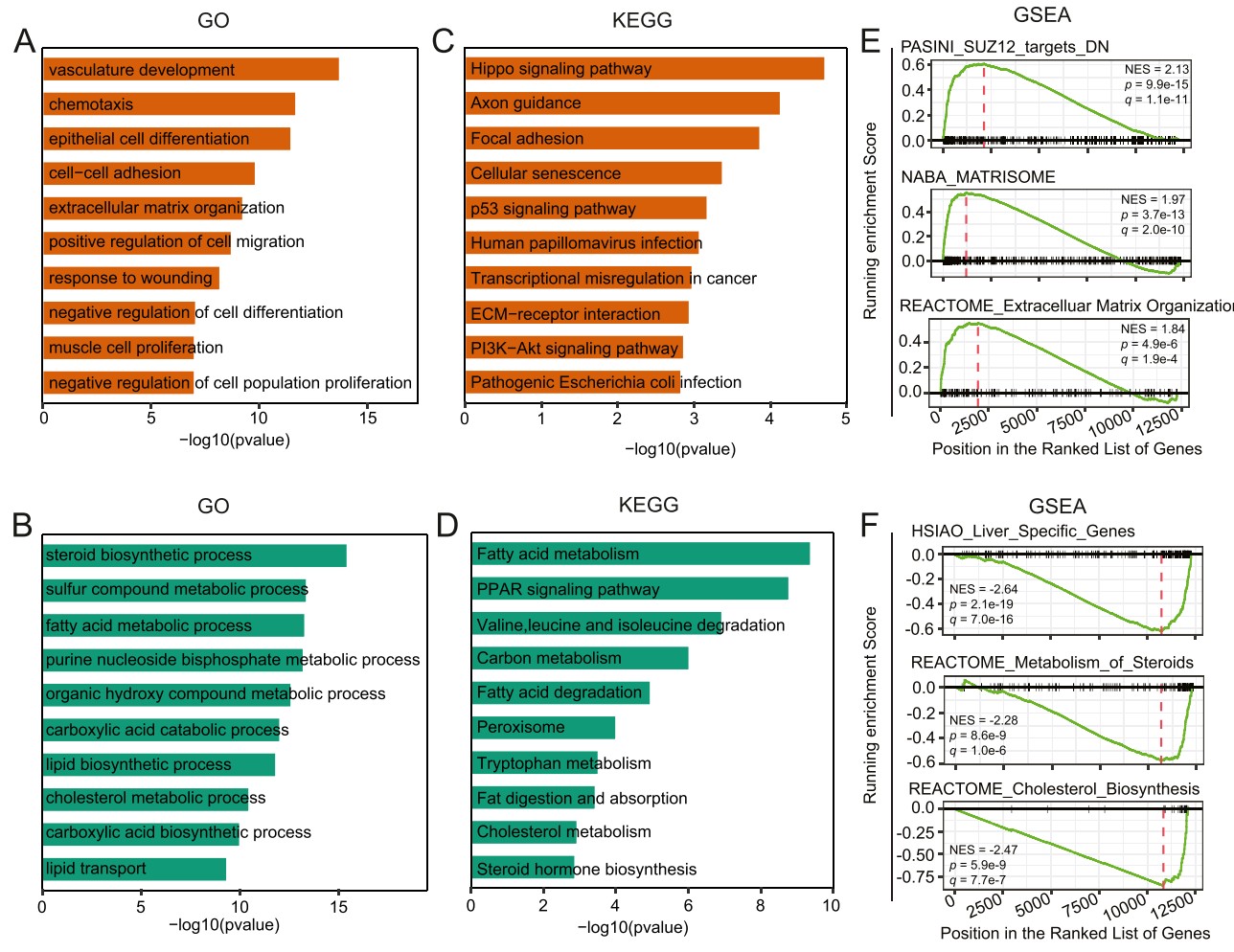

**Figure 2. Functional enrichment analysis of the differentially expressed genes induced by Yin Yang 1 knockdown.**
**(A, B)** Gene Ontology enrichment analysis for up-regulated (A) and down-regulated genes (B). **(C, D)** Kyoto Encyclopedia of Genes and Genomes pathway enrichment analysis for up-regulated (C) and down-regulated genes (D). **(E, F)** Gene Set Enrichment Analysis for up-regulated (E) and down-regulated genes (F). The C2: curated genes sets of the Molecular Signature Database (MSigDB) were used as input (Subramanian et al, 2005).

## Extensive dysregulation of hepatic lipid metabolism by YY1 knockdown

We next investigated the genes and pathways in hepatic lipid metabolism dysregulated by YY1 knockdown based on the annotations from the Reactome database (Jassal et al, 2020). We observed extensive down-regulation of genes in multiple biological processes in lipid metabolism, such as FA synthesis and transport, mitochondrial FAO and ketogenesis, peroxisomal FAO, cholesterol biosynthesis, and lipoprotein metabolism (Fig 3A). For genes involved in phospholipid and sphingolipid metabolism, similar numbers of down-regulated and up-regulated genes are present (Fig S2). We then used RT-qPCR with RNA from two batches of samples (*n* = 8; independent from the samples used in RNA-seq) as templates to validate the identified DEGs in lipid metabolism. A total of 31 DEGs encoding important enzymes or proteins in abovementioned biological processes in lipid metabolism were selected for RT-qPCR validation. We observed that except for *CPT1A*, all the other 30 DEGs showed similar degree of down-regulation in

RT-qPCR in comparison with the RNA-seq data (Fig 3B–G and Table S3). This extensive down-regulation of genes in lipid metabolism was further validated at protein level. This is illustrated by Western blot analysis on four of the down-regulated key enzymes in FA synthesis involved in both monounsaturated FA synthesis (SCD) and polyunsaturated FA synthesis (FADS1, FADS2, and ELOVL2) (Fig 3H).

## Dysregulation of multiple TFs and their coactivators in lipid metabolism by YY1 knockdown

Hepatic lipid metabolism is subject to stringent and coordinate regulations at transcriptional level by multiple TFs (Wang et al, 2015b; Piccinin et al, 2019). To find the molecular links between YY1 knockdown and dysregulated lipid metabolism, we further explored the dysregulated TFs and their coactivators or corepressors known to be involved in the regulation of lipid metabolism. Several nuclear receptors (NRs) including vitamin D receptor (VDR), liver X receptor β (LXRB), farnesoid X receptor (FXR), and peroxisome proliferator activated receptor α (PPARA), which regulate lipid metabolism by

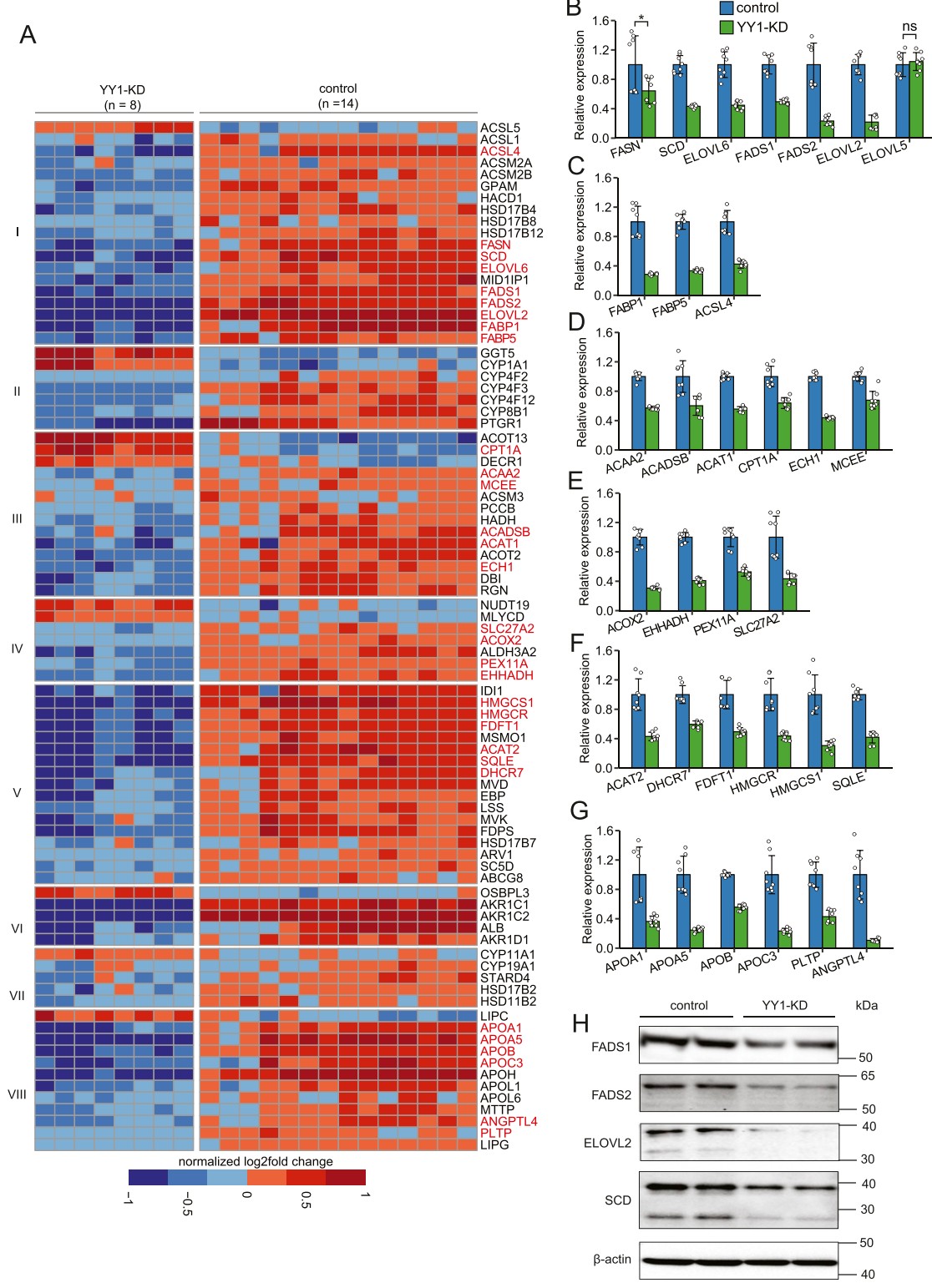

**Figure 3. Dysregulated lipid metabolism by Yin Yang 1 knockdown.**
**(A)** Heat map showing enrichment of differentially expressed genes (DEGs) in various pathways in hepatic lipid metabolism. The assignment of the DEGs to each pathway was done with the Reactome database as reference (Jassal et al, 2020). Pathways showing here include (I) fatty acid (FA) synthesis and transport; (II) arachidonic acid metabolism (III) mitochondrial FA beta-oxidation (FAO) and ketogenesis; (IV) peroxisomal FAO; (V) cholesterol biosynthesis; (VI) bile acid and bile salt metabolism; (VII) metabolism of steroid hormones; and (VIII) lipoprotein metabolism. The DEGs that are further validated by RT-qPCR are colored in red. **(B, C, D, E, F, G)** RT-qPCR validation of the DEGs in hepatic lipid metabolism. The DEGs encoding key enzymes or proteins in FA synthesis (B), FA transport (C), mitochondrial FAO and ketogeneisis

forming heterodimers with retinoid X receptor $\alpha$ (RXRA) and RXRB, were found dysregulated by YY1 knockdown in RNA-seq (Fig 4A). Considering the involvement of these heterodimers in lipid metabolism, we systematically evaluated the expression changes of the NRs and their heterodimeric partners by RT-qPCR and Western blot (Lefebvre et al, 2010). *RXRA* and *RXRB* are expressed in the liver and neither were defined as DEGs in our RNA-seq analysis; however, the expression of *RXRA* showed the trend of down-regulation in RNA-seq with $\log_2$ fold change of $-0.56$ and *P*-value of $5.5 \times 10^{-15}$, whereas *RXRB* showed no sign of down-regulation. This was validated in RT-qPCR which proved that only *RXRA* was significantly down-regulated by YY1 knockdown (Fig 4B). The down-regulation of RXRA was further validated by Western blot analysis (Fig 4E). For various NRs that heterodimer with RXRs, the expression changes of *LXRA* and *LXRB*, *PPARA*, *PPARD* and *PPARG*, *FXR*, and *VDR* were first validated with RT-qPCR (Fig 4B). We confirmed the significant down-regulation of both *FXR* and *PPARA* but not the other NRs in RT-qPCR. The observed up-regulation of *LXRB* and *VDR* by YY1 knockdown in RNA-seq could not be replicated in our RT-qPCR analysis. We then used Western blot analysis to verify the decreased expression of FXR and PPARA at protein level. In accordance with results from both RNA-seq and RT-qPCR, the expression of FXR and PPARA was also observed significantly down-regulated by YY1 knockdown in Western blot (Fig 4E). For PPARA, human 293T cells overexpressing *PPARA* were used as the positive control which led to a significant increase in expression at 52 kD (Fig 4F). Knocking down of *YY1* caused the significant down-regulation of the activated 59-kD form of PPARA but not the 52-kD form (Fig 4E) (Passilly et al, 1999).

The NR complexes exert their regulatory effects by binding to specific DNA-binding sites and recruiting multiple coactivators or corepressors. In addition to *FXR* and *PPARA*, several coactivators of the NR complexes including PPARG coactivator 1 $\alpha$ (*PGC1A*), *PGC1B*, NR coactivator 1 (*NCOA1*), *NCOA2*, and *NCOA3* were significantly down-regulated by YY1 knockdown (Fig 4A). The down-regulations of these coactivators were confirmed in following RT-qPCR analysis with PGC1A also verified in Western blot analysis (Fig 4C and E).

Knocking down of *YY1* caused extensive down-regulation of genes involved in hepatic lipogenesis pathway, especially genes involved in FA synthesis (Fig 3A and B). Sterol regulatory element binding protein 1 (SREBP1) and carbohydrate responsive element binding protein (CHREBP) are two of the major TFs regulating hepatic lipogenesis (Xu et al, 2013; Wang et al, 2015b). Our RNA-seq analysis revealed significant down-regulation of *CHREBP* but not sterol regulatory element binding transcription factor 1 (*SREBF1*), which encoding SREBP1. The expression changes of *SREBF1* together with *CHREBP* and its heterodimeric partner MAX dimerization protein (*MLX*) were then verified by RT-qPCR which confirmed that only *CHREBP* was significantly down-regulated (Fig 4D). Western blot analysis with antibodies against CHREBP and SREBP1, respectively, also confirmed that only CHREBP was significantly down-regulated by YY1 knockdown (Fig 4E). Knocking down of *YY1* also caused significant

down-regulation of almost all the key enzymes in cholesterol biosynthesis (Fig 3A). The expression of *SREBF2*, encoding SREBP2, was significantly down-regulated by YY1 knockdown in RNA-seq (Fig 4A). The down-regulation of *SREBF2* was further validated by both RT-qPCR and Western blot (Fig 4D and E).

Beside the abovementioned TFs and theirs coactivators, the hepatocyte nuclear factors (HNFs) HNF4A, forkhead box A1 (FOXA1), and forkhead box A2 (FOXA2) are important transcription regulators of both liver organ development and liver metabolism (Lau et al, 2018). *FOXA1* was identified as one of the DEGs down-regulated by YY1 knockdown in RNA-seq (Fig 4A). RT-qPCR and Western blot analysis confirmed that not only *FOXA1* but all the three HNFs were significantly down-regulated by YY1 knockdown (Fig 4D and E).

### Intersection between YY1 ChIP-seq and RNA-seq data unravels candidate molecular mechanisms regulating the expression of the identified DEGs

The presence of YY1 binding nearby the DEGs can be indicative of a direct regulatory role of YY1 in regulating the DEGs expression. To achieve better understanding of the molecular mechanisms regulating the transcriptional alternations induced by YY1 knockdown, we obtained YY1 chromatin immunoprecipitation coupled with massively parallel sequencing (ChIP-seq) data in HepG2 cells and liver tissues from the ENCODE project (The Encode Project Consortium, 2012). In accordance with previous observations that YY1 commonly occupied active promoters and enhancers, the reads in YY1 ChIP-seq experiments from both HepG2 cells and liver tissues were highly enriched in the promoter regions (Fig 5A and B) (Weintraub et al, 2017). By intersecting the DEGs with their nearby YY1-binding peaks defined in HepG2 cells, we identified 596 down-regulated genes bearing 902 YY1 peaks nearby and 620 up-regulated genes bearing 933 YY1 peaks nearby (Fig 5C). We further divided the DEGs based on their overall deregulation and the presence of YY1 peaks. Fig 5D shows that of the DEGs that have YY1 bound nearby, which are presumed to be the direct targets of YY1, approximately equal proportions of the up-regulated and down-regulated genes are evenly distributed based on their changes in expression level. A larger fraction of the up-regulated genes did not have YY1 bound nearby, especially for the DEGs that are up-regulated more than fourfold by YY1 knockdown.

To find the causal TFs that are enriched over the DEGs with or without YY1 bound nearby, we further performed enrichment analysis against the ChIP-X enrichment analysis (ChEA) gene set library with the Enrichr webtool (Kuleshov et al, 2016). The results showed that the up-regulated genes with YY1 bound nearby were significantly enriched with CTCF and the cohesin complex components SMC3 and RAD21 binding, whereas the up-regulated genes without YY1 bound nearby were significantly enriched with binding of SUZ12, a subunit of the PRC2 (Fig 5E). CTCF and cohesin usually work as a complex that binds to the same genomic regions to organize higher order chromatin structures and regulate gene

(D), peroxisomal FAO (E), cholesterol biosynthesis (F), and lipoprotein metabolism (G) were selected for RT-qPCR validation. Error bars, SD; *n* = 8 technical replicates. \*\**P* < 0.01 as the default; \**P* < 0.05 and ns, not significant calculated by two-tailed *t* tests. The detailed *P*-value for each gene is listed in Table S3. **(H)** Western blot analysis confirmed the significant down-regulation of key enzymes in FA synthesis by Yin Yang 1 knockdown. The expression of $\beta$-actin was used as the loading control.

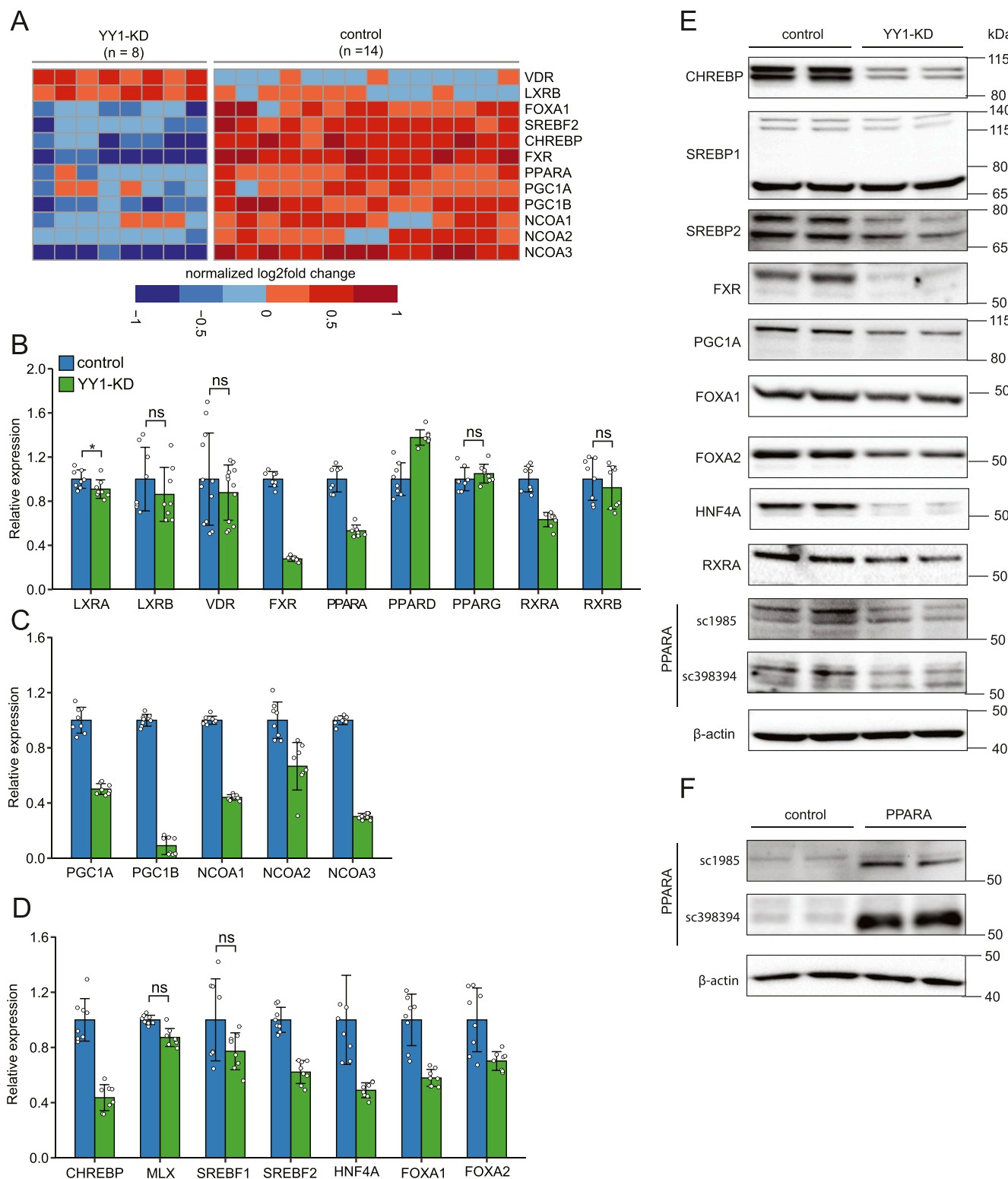

**Figure 4.   Multiple transcription factors and coactivators that are important for hepatic lipid metabolism were down-regulated by Yin Yang 1 (YY1) knockdown.**
**(A)** Heat map of the differentially expressed genes encoding important transcription factors and their coactivators in hepatic lipid metabolism induced by YY1 knockdown. **(B, C, D)** RT-qPCR validation of the expression changes of various nuclear receptors in heterodimer with RXR (B) together with their transcriptional coactivators (C) induced by YY1 knockdown. The expression changes of other key transcription factors involved in hepatic lipid metabolism were also validated (D). Error bars, SD; $n$ = 8 technical replicates ($n$ = 12 for $VDR$). **P < 0.01 as the default; *P < 0.05 and ns, not significant calculated by two-tailed $t$ tests. The detailed $P$-value for each gene is listed in Table S3. **(E, F)** The down-regulation of the identified key transcription factors and coactivators by YY1 knockdown was further confirmed by Western blot

expression (Fig 5F) (Wutz et al, 2017). As the expression of both CTCF and cohesin components was not affected by YY1 knockdown, the up-regulation of nearby genes may be caused by disruption of the local promoter and enhancer interactions mediated by YY1 and cohesin (Weintraub et al, 2017). The down-regulated genes with YY1 bound nearby were significantly enriched with binding of HNF4A, LXR, FOXA2, RXR, and PPARA, which are all important TFs in lipid metabolism, whereas the down-regulated genes without YY1 bound nearby were enriched with binding of HNF4A only (Fig 5G). RXRA and HNF4A were observed to frequently bind to the same regulatory regions in ChIP-seq experiments in HepG2 cells (Fig 5H). The down-regulation of RXRA and its heterodimeric partner PPARA together with the down-regulation of HNF4A and YY1 may have caused the down-regulation of nearby genes. To further verify our hypothesis, we restrict our analysis on the subset of 137 DEGs composed of all the identified genes and TFs involved in hepatic lipid metabolism (Figs 3A, 4A, and S2). Among the 137 DEGs, 94 of them had YY1 peaks nearby which were significantly enriched with binding of RXR, PPARA, LXR, HNF4A, FOXA2, and PPARG. For 74 of the 94 DEGs, at least one of the abovementioned TFs together with YY1 were bound nearby. For the 43 DEGs without YY1 peaks nearby, they were only significantly enriched with binding of FOXA2 (Fig 5I).

### YY1 regulates PPAR signaling pathway by targeting PPARA

PPARs regulate target gene expression by forming heterodimer with RXR and binding at specific DNA response elements called PPAR response elements (PPREs) (Dubois et al, 2017). The expression of *PPARA* and *RXRA* together with several of their coactivators including *PGC1A*, *PGC1B*, and *NCOA1/2/3* were significantly down-regulated by YY1 knockdown (Fig 4). This suggested that decreased expression of these factors changed the regulatory activities of PPREs which further led to significant down-regulation of nearby target genes. This is illustrated by the PPAR signaling pathway (KEGG: hsa03320) being one of the most significantly down-regulated pathways induced by YY1 knockdown (Fig 2D).

We next sought to determine how *PPARA* was down-regulated by YY1 knockdown. In both HepG2 cells and liver tissues, we found multiple YY1 ChIP-seq peaks at the promoter (defined as P) and upstream enhancer regions (defined as E1–E4) of *PPARA* (Fig 6A). The binding of YY1 at these regions was further validated with chromatin immunoprecipitation followed by quantitative real-time PCR (ChIP-qPCR) with antibody against YY1 in HepG2 cells (Fig 6B). The promoter region of *PPARA* showed strong promoter activity in luciferase assay. Upon *YY1* overexpression, the promoter activity was significantly induced (Fig 6C). All the YY1-binding enhancer regions showed strong enhancer activities compared with the control pGL4.23 plasmid in luciferase assay. The enhancer activities of all the four enhancers were also significantly induced by *YY1* overexpression (Fig 6D). This shows that YY1 directly regulates *PPARA* expression by binding to its promoter and upstream enhancer regions.

In addition to *PPARA*, the expression of *FABP1* was also significantly down-regulated by YY1 knockdown. As a well-known target gene regulated by PPARA, the protein encoded by *FABP1* functions as a mandatory vehicle in the liver for transport of PPARA agonists into the nucleus to activate the PPARA/RXRA complex for target gene trans-activation (Guzmán et al, 2013; Wang et al, 2015a). YY1 was found to bind to the promoter region of *FABP1* overlapping with the previously identified PPREs by exploring the ChIP-seq signals of YY1 and RXRA in HepG2 cells (Fig 6E) (Guzmán et al, 2013). The enriched binding of both YY1 and PPARA/RXRA complex to the promoter region of FABP1 was further verified with ChIP-qPCR with antibodies against YY1, PPARA, and RXRA, respectively (Fig 6F). To verify if the promoter of *FABP1* responds to YY1 overexpression, we constructed two luciferase constructs FP1 and FP2 to test their promoter activities. Both of the promoter constructs showed strong promoter activities compared with pGL4.10 in luciferase assay (Fig 6G). In accordance with previous observations, the promoter activity of the *FABP1* promoter region was significantly induced by *PPARA* overexpression but not by *YY1* overexpression (Fig 6H and I). The expression of *FABP1* is reported to be directly regulated by PPARA, FOXA1, and HNF4A, which were all down-regulated by YY1 knockdown (Fig 4) (Guzmán et al, 2013). This strongly suggests that the down-regulation of *FABP1* induced by YY1 knockdown was directly caused by the decreased expression of PPARA, FOXA1, and HNF4A, and indirectly caused by YY1. Altogether, our data support the idea that knocking down of *YY1* significantly down-regulates the expression of several key components of the PPARA/RXRA complex, which further affected the expression of its downstream genes.

### The DEGs in FA synthesis are regulated by the cooperation between YY1 and PPARA

The enriched binding of YY1, RXR, PPARA, and PPARG nearby the DEGs in hepatic lipid metabolism suggested that YY1 and the PPAR/RXR complex may cooperate with each other in regulating target gene expression (Fig 5I). The down-regulation of both YY1 and components of the PPAR/RXR complex may all contributed to the decreased expression of the DEGs with these TFs bound nearby. This cooperation was not observed at the promoter region of *FABP1* as it only responded to *PPARA* overexpression (Fig 6H and I). However, we observed that YY1 peaks frequently overlapped with RXRA peaks at the promoter regions of the DEGs in the PPAR signaling pathway (KEGG: hsa03320) in HepG2 cells. As the expression of YY1 and PPARA/RXRA were all down-regulated by YY1 knockdown, we next evaluated the cooperation between YY1 and PPARA/RXRA in regulating the target gene expression. We focused our analysis on the genes involved in FA synthesis pathways including both genes encoding key enzymes in mono-unsaturated (*FASN*, *SCD*, and *ELOVL6*) and polyunsaturated FAs synthesis (*FADS1*, *FADS2*, *ELOVL2*, and *ELOVL5*). First, except for *ELOVL5*, all the other six genes were significantly down-regulated by YY1 knockdown validated by both RT-qPCR and Western blot analysis (Fig 3B and H). Second, both YY1 and RXRA peaks were

analysis (E). The expression of peroxisome proliferator activated receptor α (PPARA) was detected with two different antibodies SC-1985 and SC-398394 from Santa Cruz Biotechnology. (F) Western blot analysis of PPARA in human 293T cells overexpressed with PPARA was used as the positive control (F). The 125-kD precursor of SREBP2 was not detected in our analysis, which was not shown here. The expression of β-actin was used as the loading control in both HepG2 and 293T cells.

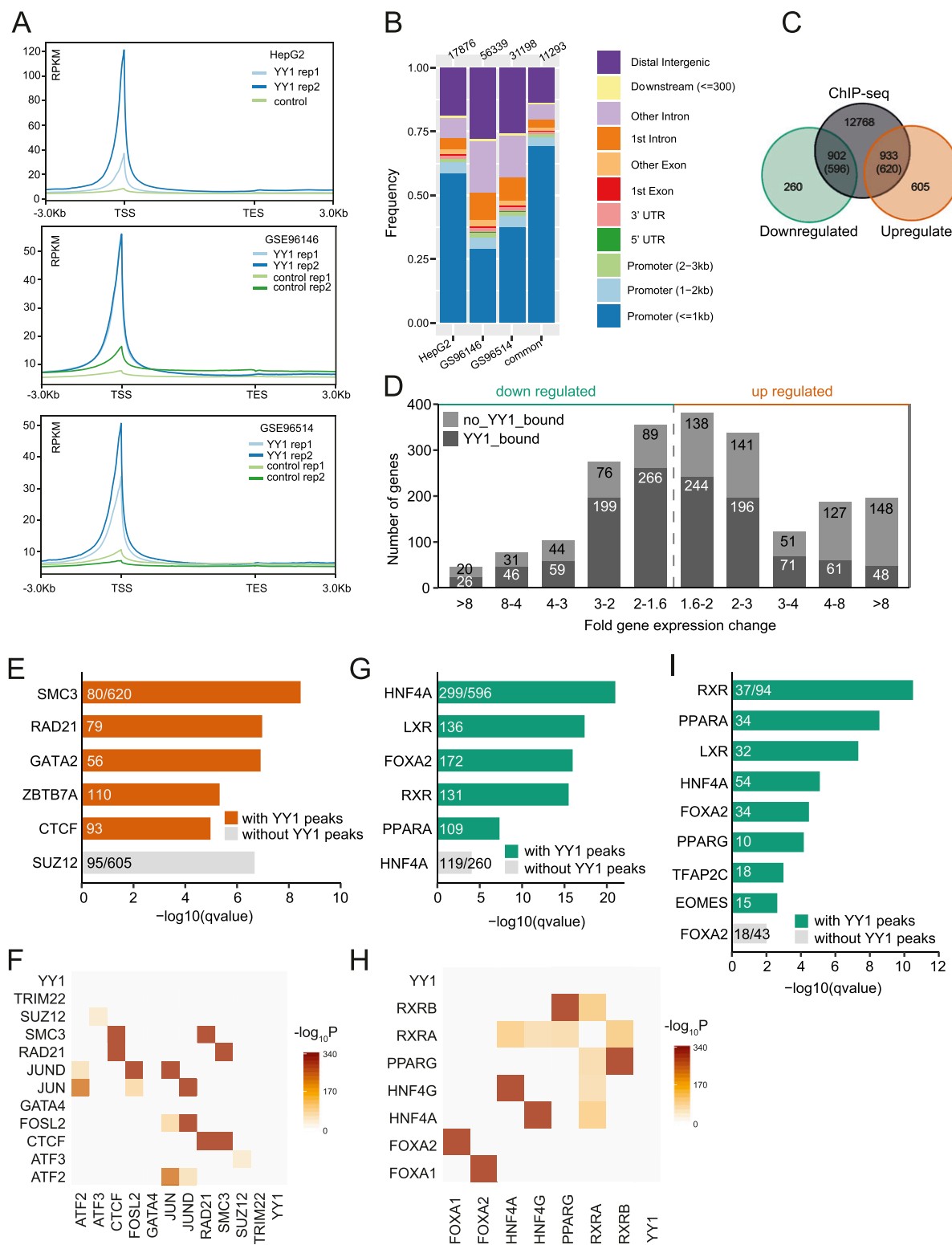

**Figure 5. Intersection between Yin Yang 1 (YY1) ChIP-seq and RNA-seq data unravels molecular mechanisms regulating target gene expression.**
**(A)** The reads distribution of YY1 ChIP-seq data over gene bodies in liver tissues and HepG2 cells. **(B)** Annotation of YY1 peaks based on their genomic locations. **(C)** Intersection between YY1 ChIP-seq peaks and differentially expressed genes (DEGs) induced by YY1 knockdown. The YY1 peaks that were annotated as distal intergenic were disregarded in this analysis. **(D)** Histogram of fold gene expression change for YY1-bound (dark grey) and unbound DEGs (light grey). **(E)** Enrichment of transcription factors (TFs) binding nearby up-regulated genes induced by YY1 knockdown. The consensus gene sets from the ENCODE project and ChEA database were used as the input (Kuleshov et al, 2016). **(F)** The enriched TFs bound nearby up-regulated genes were significantly co-occurred in the same regulatory regions determined by tfNet

enriched over the promoter regions of these genes. Third, previous studies proved that several genes in FA synthesis including *FADS2*, *SCD*, and *MOD1* were directly regulated by PPARA through binding to the PPREs in their promoters in different species (Miller & Ntambi, 1996; Tang et al, 2003; Rakhshandehroo et al, 2010). Last, even though PPARA contributes to lipid metabolism by majorly targeting the FAO pathway, new evidence suggested that PPARA is also involved in FA synthesis pathways (Rakhshandehroo et al, 2009; Burri et al, 2010; Pawlak et al, 2015; Dubois et al, 2017). As many of the unsaturated FAs are important PPARA agonists, the activation of the FA synthesis pathways by PPARA was suggested to ensure enough FAs agonists for its own activation (Burri et al, 2010).

We first evaluated both the promoter and enhancer regions bound by YY1 and/or RXRA nearby the selected genes, except for *ELOVL5* the expression of which was unchanged by *YY1* knockdown (Fig 3B). For the six DEGs, both YY1 and RXRA peaks were observed to co-occur at their promoter regions. The *ELOVL2* locus also had four enhancer regions bound by RXRA and the *SCD* locus has two enhancer regions indicative of RXRA binding (Fig S3A and E). All the promoter regions had strong promoter activities in luciferase assay (Fig 7A). The promoter activities of the selected genes were significantly induced by *YY1* overexpression except for *FASN* (Fig 7B). Except for the promoter region of *FADS1*, the promoter regions of the selected genes were also highly responsive to *PPARA* overexpression (Fig 7C). For the identified enhancer regions in the *ELOVL2* and *SCD* loci enriched with RXRA binding, five out of the six enhancer regions showed significant enhancer activity in luciferase assay (Fig S3B and F). Out of the six enhancer regions, only one region in the *SCD* locus (named SCD_en2) showed significant response to PPARA overexpression in luciferase assay (Fig S3C, D, G, and H). We compared the promoter activities induced by either *YY1* or *PPARA* overexpression against *YY1* and *PPARA* overexpression together in luciferase assay. The results showed that the promoter activities of all the four promoter regions (*FADS2*, *ELOVL2*, *ELOVL6*, and *SCD*) were significantly induced by the co-overexpression of *YY1* and *PPARA* compared with overexpression of only *YY1* or *PPARA* (Fig 7D). The promoter regions of *FADS2* and *ELOVL2* failed to be induced by *PPARA* overexpression under this experimental set up. This is probably due to the relatively weak responses of these two promoters to *PPARA* overexpression compared with the promoter regions of *ELOVL6* and *SCD* (Fig 7C). The trend is more noticeable for the promoter regions of both *ELOVL6* and *SCD*, which showed significant responses to both single- and co-overexpression of *YY1* and *PPARA* compared with the control. Both the promoter regions of *ELOVL6* and *SCD* were then selected for further study on how their activities were regulated by the co-operation between YY1 and PPARA.

### Fine mapping of the YY1- and PPARA-responsive regions in the *SCD* promoter

In accordance with the promoter region of *SCD* responding to both *YY1* and *PPARA* overexpression in luciferase assay, this region is enriched with both YY1 and RXRA binding. The initial luciferase construct of the *SCD* promoter encompassed a region ranging from +6 to −1,289 relative to the translational start site of *SCD* that totally covered the overlapping region bound by both YY1 and RXRA (Fig 8A). To delineate the minimal regions responsible for the transcriptional regulation by YY1 and PPARA/RXRA, several truncation constructs of the *SCD* promoter were generated (Fig 8B). The minimal region responsible for the constitutive transcription activity of *SCD* lays within the region ranging from −272 to −640 (Fig 8B). The largest difference in luciferase activity was found between the −448 and the −640 constructs indicating the 192-bp region ranging from −448 to −640 to be the key regulatory region for the *SCD* promoter. These truncation constructs were further subjected to either *YY1* or *PPARA* overexpression. The minimal region that responds to *YY1* overexpression was mapped to the region spanning from +6 to −640 demonstrated by the gradual increase in the luciferase activity of luciferase constructs −272, −448, and −640 induced by YY1 (Fig 8C). The minimal region that responds to *PPARA* overexpression was mapped to the region ranging from −448 to −640 (Fig 8D).

In accordance with the ChIP-seq data, the *SCD* promoter was enriched with binding of both YY1 and PPARA/RXRA validated by ChIP-qPCR in HepG2 cells (Fig 8E). We then used the TRAP tool to search for putative binding sites for YY1 and PPARA/RXRA (Thomas-Chollier et al, 2011). There are four conserved YY1-binding sites and two PPREs predicted (Fig 8A). In accordance with the PPARA responding region located in the region from −448 to −640, the two PPREs were predicted to be centrally located at −497 and −599, respectively. The YY1-binding sites were predicted to be located at +2, −362, −422 and −497, respectively. The predicted YY1-binding site at −497 was also located in one of the PPREs and was named PPA/YY1 (−497). Fig 8F shows the sequences of the predicted YY1-binding sites, whereas Fig 8G shows the sequences of the two predicted PPREs presumed to be bound by PPARA/RXRA. We further mutated the predicted YY1-binding sites and the two PPREs in the *SCD* promoter to test the changes in response to both *YY1* and *PPARA* overexpression in luciferase assay (Fig 8F and G). Introducing mutations to both YY1 (−422) and PPA/YY1 (−497) significantly decreased the promoter activity (Fig 8H). Accordingly, the response of the *SCD* promoter to *YY1* overexpression was significantly compromised by mutations introduced to these two sites (Fig 8I). Introducing mutations to either of the predicted PPREs or one of the YY1-binding site YY1 (−422) significantly decreased the response to *PPARA* overexpression in luciferase assay (Fig 8J). In accordance with the expression of mouse *SCD* regulated by PPARA binding to the PPREs in its promoter region, the identification of functional PPREs in the human *SCD* promoter suggests that the PPREs in *SCD* promoter are evolutionarily conserved between human and mouse (Miller & Ntambi, 1996). Altogether, our complementary results proved that the region spanning from −448 to −640 is the key regulatory region for the *SCD* promoter which contains key binding sites for both YY1 and PPARA/RXRA.

---

(Diamanti et al, 2016). **(G)** Enrichment of TFs binding nearby down-regulated genes induced by YY1 knockdown. The ChEA database was used as the input (Kuleshov et al, 2016). **(H)** The enriched TFs bound nearby down-regulated genes were significantly co-occurred in the same regulatory regions determined by tfNet (Diamanti et al, 2016). **(I)** Enrichment of TFs binding nearby the DEGs involved in hepatic lipid metabolism. The ChEA database was used as the input (Kuleshov et al, 2016).

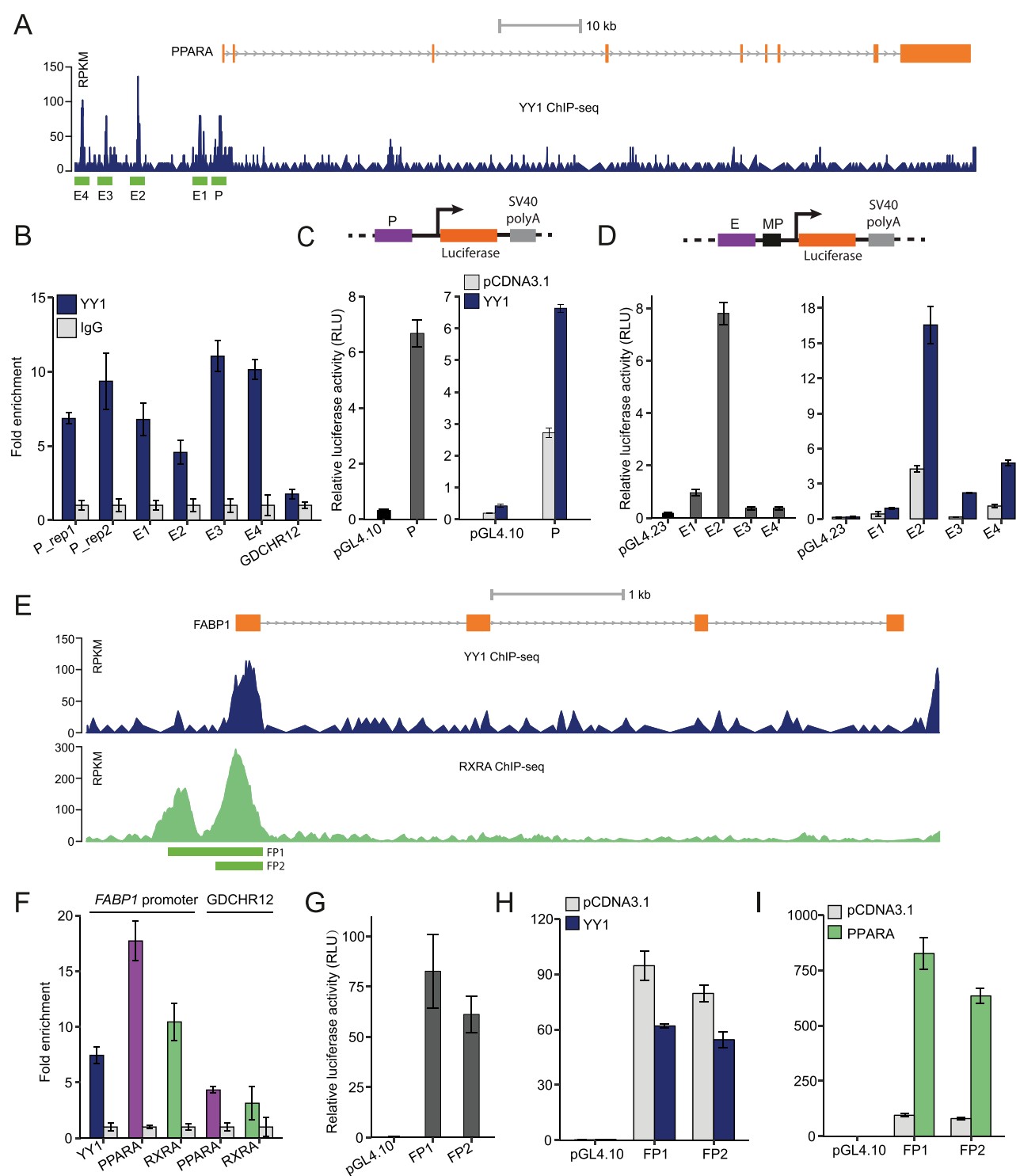

**Figure 6. Yin Yang 1 (YY1) regulates peroxisome proliferator activated receptor α (PPARA) expression by binding to its promoter and upstream enhancer regions.**
**(A)** The ChIP-seq signals of YY1 in the *PPARA* locus in HepG2 cells. The defined YY1-binding peaks were highlighted in green and named as P (promoter) and E1–E4 (enhancer) based on their location relative to PPARA. **(B)** ChIP-qPCR validation of YY1 binding over the identified peak regions in the *PPARA* locus in ChIP-seq. Fold change of YY1 chromatin binding compared with IgG control is presented for each region with GDCHR12 used as the negative control. Error bars, SD; *n* = 3 technical replicates. **(C)** The promoter region of *PPARA* showed strong promoter activity in luciferase assay. Upon *YY1* overexpression, the promoter activity was significantly induced when compared with the activity of the control pGL4.10 plasmid. **(D)** The identified enhancer regions bound by YY1 in the *PPARA* locus (E1–E4) showed enhancer activities which were significantly induced by *YY1* overexpression in luciferase assay. **(E)** Both YY1 and RXRA bound to the promoter region of *FABP1* identified from ChIP-seq signals of YY1 and RXRA in HepG2 cells. **(F)** ChIP-qPCR validation of YY1, PPARA, and RXRA binding to the promoter region of *FABP1*. Fold change of transcription factors chromatin binding compared with IgG control is presented with GDCHR12 used as the negative control. Error bars, SD; *n* = 3 technical replicates. **(G, H, I)** The promoter region of *FABP1*

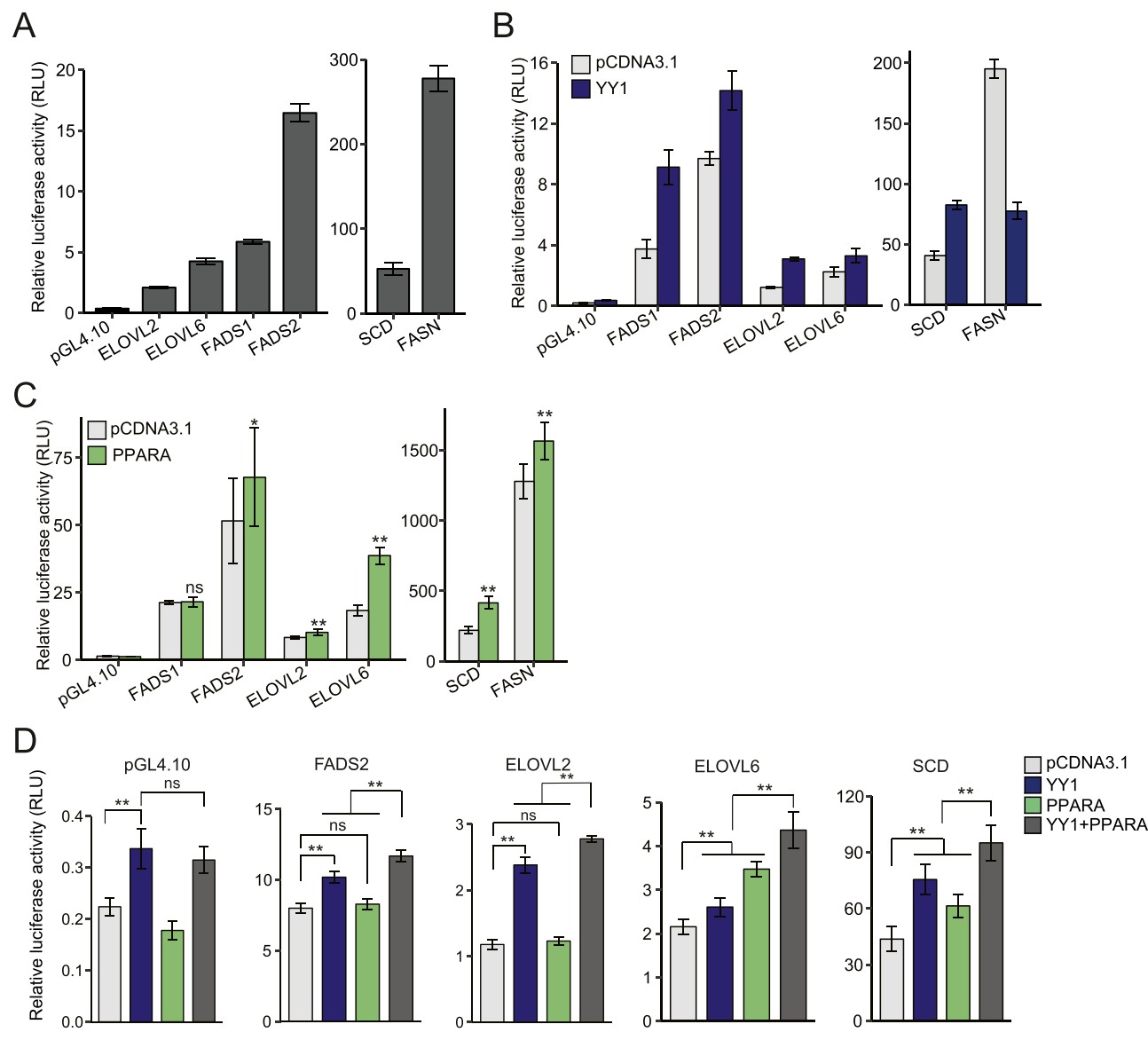

**Figure 7. The cooperation between Yin Yang 1 (YY1) and peroxisome proliferator activated receptor α (PPARA) determined the promoter activities of genes encoding key enzymes in fatty acid (FA) synthesis.**
**(A, B, C)** The promoters of genes encoding key enzymes in FA synthesis showed strong promoter activities in luciferase assay (A). **(B, C)** The promoters also highly responded to both YY1 overexpression (B) and PPARA overexpression (C) in luciferase assay. **(D)** Overexpression of *YY1* together with *PPARA* significantly increased the promoter activities of genes encoding key enzymes in FA synthesis compared with overexpression of *YY1* only or *PPARA* only. For (A, B, C, D), mean ± SD of six technical replicates from two independent plasmid extractions and transfections with each transfection had three technical replicates. **$P < 0.01$ and ns, not significant calculated by two-tailed *t* tests.

## Fine mapping of the YY1- and PPARA-responsive regions in the *ELOVL6* promoter

The promoter region of *ELOVL6* was also enriched with both YY1 and RXRA binding. The luciferase construct of the *ELOVL6* promoter contains the region ranging from −285 to −894 relative to the translational start site which covered the overlapping region bound by both YY1 and RXRA (Fig 9A). To identify the key regions responding to *YY1* and/or *PPARA* overexpression, we created a series of truncation constructs (Fig 9B). The minimal region responsible for the basal promoter activity was mapped to the region from −285 to −503 (Fig 9B). By subjecting to either *YY1* or *PPARA*

(luciferase constructs FP1 and FP2) showed strong promoter activity (G) which was highly induced by *PPARA* overexpression (I) but not by *YY1* overexpression (H). For (C, D, G, H, I), mean ± SD of six technical replicates from two independent plasmid extractions and transfections with each transfection had three technical replicates.

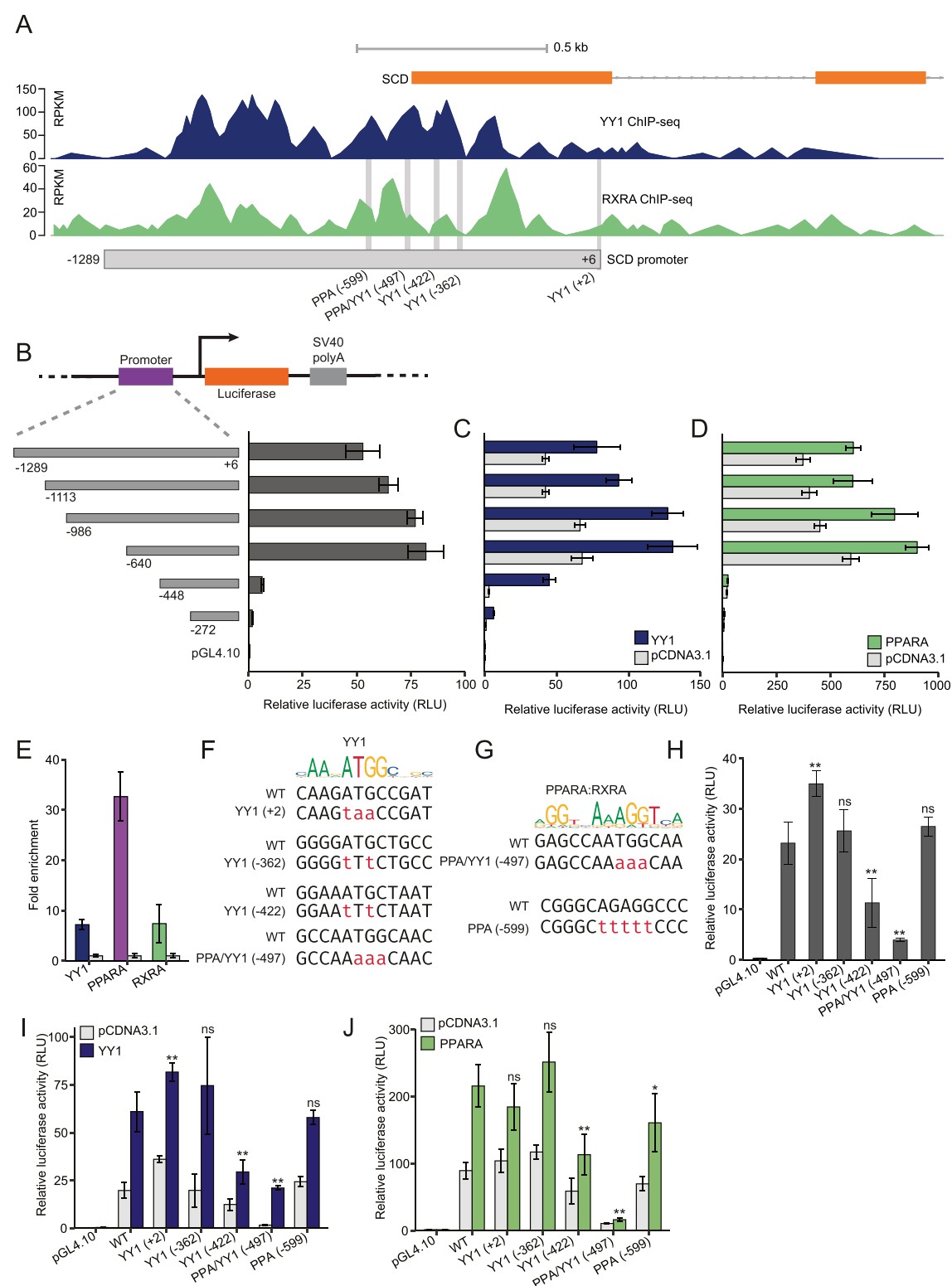

**Figure 8. Fine mapping and validation of the key genomic regions responding to Yin Yang 1 (*YY1*) and peroxisome proliferator activated receptor α (*PPARA*) overexpression in the *SCD* promoter.**
**(A)** The YY1 and RXRA ChIP-seq signals in the promoter region of *SCD* in HepG2 cells. The light grey box demonstrates the location of the promoter region (−1,289 to +6) examined in luciferase assay. The vertical light grey lines demonstrate the predicted binding sites for YY1 and PPARA/RXRA. **(B)** The minimal promoter region of *SCD* is fine mapped to a ~600-bp region (luciferase construct −640 relative to the translational start site of *SCD*) by luciferase assay with a series of truncation constructs. **(C, D)** The key regions of the *SCD* promoter respond to either *YY1* (C) or *PPARA* (D) overexpression were finely mapped to the minimal promoter by luciferase assay with the same set

overexpression, the minimal region responding to *YY1* over-expression was mapped from −503 to −702, whereas the PPARA responsive region was mapped from −381 to −503 (Fig 9C and D). In accordance with the ChIP-seq data, the *ELOVL6* promoter was not only enriched with binding of previously identified YY1 and RXRA but also highly enriched with PPARA binding validated by ChIP-qPCR in HepG2 cells (Fig 9E). There are two conserved YY1-binding sites and one conserved PPREs predicted in the *ELOVL6* promoter by the TRAP tool (Fig 9A) (Thomas-Chollier et al, 2011). The two YY1-binding sites were predicted to be located at −593 and −618, respectively, which is in accordance with the identified YY1-responsive region (Fig 9F). The predicted PPREs was located at −478, which is also in accordance with the identified PPARA-responding region (Fig 9G). We then introduced mutations to the two predicted YY1-binding sites and the PPRE in the original *ELOVL6* promoter luciferase construct to test the changes in response to both *YY1* and *PPARA* overexpression in luciferase assay (Fig 9F and G). The *ELOVL6* promoter activity was not decreased by the mutations introduced to either the YY1-binding sites or the PPRE in luciferase assay (Fig 9H). Similarly, introducing mutations to both the YY1-binding sites and the PPRE in the *ELOVL6* promoter did not significantly decrease its response to *YY1* overexpression (Fig 9I). As there are no other YY1-binding sites predicted in the *ELOVL6* promoter, the response of the *ELOVL6* promoter to *YY1* over-expression might be mediated either by some non-canonical YY1-binding sites or indirectly by other cofactors in complex with YY1 (Zhang et al, 2017). In accordance with our prediction, introducing mutations to the predicted PPRE significantly decreased the promoter activity of the *ELOVL6* promoter in response to *PPARA* overexpression (Fig 9J). This suggested that the predicted PPREs located at −478 in the *ELOVL6* promoter is essential for PPARA/RXRA binding, which further determines its promoter activity.

## Discussion

We here demonstrated that YY1 is tightly involved in the regulation of hepatic lipid metabolism by directly or indirectly regulating the expression of several key TFs and their coactivators. Knocking down of *YY1* significantly decreased the expression of CHREBP, SREBF2, FXR, multiple components of the PPARA/RXRA complex, and HNF family members including HNF4A, FOXA1, and FOXA2. Decreased expression of these TFs changed the regulatory activities of the regulatory regions bound by these TFs which further significantly decreased the expression of genes involved in lipogenesis, FAO, cholesterol biosynthesis, and lipoprotein metabolism (Fig 10).

In this study, the expression of *YY1* was significantly down-regulated by lentiviral-mediated shRNA and amiRNA silencing (Fig 1A and B). Interestingly, the up-regulated genes induced by YY1

knockdown which also had YY1-binding peaks nearby their gene bodies were significantly enriched with CTCF and cohesin complex binding nearby (Fig 5E). YY1 has been well characterized as a structural regulator mediating the promoter and enhancer interactions by cooperating with the cohesin complex (Merkenschlager & Odom, 2013; Weintraub et al, 2017). The disrupted expression of YY1 may have compromised the normal interactions between promoter and enhancers mediated by YY1, which further caused aberrant gene expression inside the topologically associating domains defined by CTCF. For genes up-regulated by YY1 knockdown but without YY1 peaks nearby, binding of SUZ12 was significantly enriched (Fig 5E). This is in accordance with the observation that the up-regulated genes induced by YY1 knockdown were significantly enriched with the target genes that are positively regulated by SUZ12 in GSEA analysis (Fig 2E). As key subunits of the PRC2, the expression switching of EZH2 to EZH1 can enable the formation of a non-canonical PRC2 complex composed of SUZ12 and EZH1 and positively regulate gene expression (Xu et al, 2015). Even though all the genes encoding subunits of the PRC2 were not defined as DEGs in our RNA-seq analysis, we did observe a trend of expression switching from *EZH2* (log$_2$ fold change −0.36 and *P*-value of 5.4 × 10$^{-5}$) to *EZH1* (log$_2$ fold change 0.33 and *P*-value of 0.03) in RNA-seq.

Dysregulated *YY1* has been reported to lead to dysregulated cellular metabolism including glycometabolism, lipid metabolism, and bile acid metabolism in the liver, which is believed to be one of the underlying mechanisms of how YY1 is involved in various liver diseases (Lu et al, 2013; Verdeguer et al, 2014; Zhang et al, 2017). In accordance with previous observations, our global RNA-seq confirmed that the down-regulated genes induced by YY1 knockdown were most significantly enriched in various biological processes in lipid metabolism (Fig 3). This broad and significant down-regulation of genes in the lipid metabolism is not only caused by the direct down-regulation of *YY1* but also by the down-regulation of several key TFs and their coactivators in lipid metabolism (Fig 4). Around half of the down-regulated genes (418 of 856) had HNF4A-binding peaks nearby (Fig 5G). As a member of the HNFs, HNF4A works as an important TF for both liver organ development and liver metabolism (Lau et al, 2018). Down-regulation of *HNF4A* together with other HNFs including *FOXA1* and *FOXA2* contributed to the down-regulation of genes in hepatic lipid metabolism demonstrated by the observation that a larger fraction of the DEGs in lipid metabolism were enriched with both HNF4A and FOXA2 binding (Fig 5I). In addition, in accordance with their biological functions, the down-regulation of these three HNFs may also contributed to the down-regulation of liver organ-specific genes observed in the GSEA analysis (Fig 2F).

Besides HNFs, TFs in the PPAR and RXR complexes were the most significantly enriched transcriptional regulators binding nearby the down-regulated genes (Fig 5G and I). Among the three isotypes of *PPARs*, *PPARA* is the main isotype expressed in the liver and is found

---

of truncation constructs. **(E)** ChIP-qPCR validation of YY1, PPARA and RXRA binding to the *SCD* promoter region. Fold changes of the target transcription factors chromatin binding compared to IgG control are presented with GDCHR12 in Fig 6B and F used as the negative control. Error bars, SD; *n* = 3 technical replicates. **(F, G)** Detailed view of the predicted binding sites for YY1 (F) and PPARA/RXRA (G) in the promoter of *SCD*. The detailed mutations introduced to the predicted transcription factor binding sties in luciferase constructs are highlighted in red. **(H)** Introducing mutations into the predicted YY1-binding site YY1 (−422) and the site bound by both YY1 and PPARA PPA/YY1 (−497) significantly decreased the promoter activity of *SCD* in luciferase assay. **(I, J)** The responses of the *SCD* promoter to both *YY1* (I) and *PPARA* (J) overexpression were significantly attenuated by the mutations introduced. For (A, B, C, H, I, J), mean ± SD of six technical replicates from two independent plasmid extractions and transfections with each transfection had three technical replicates. **P* < 0.01 and ns, not significant calculated by two-tailed *t* tests.

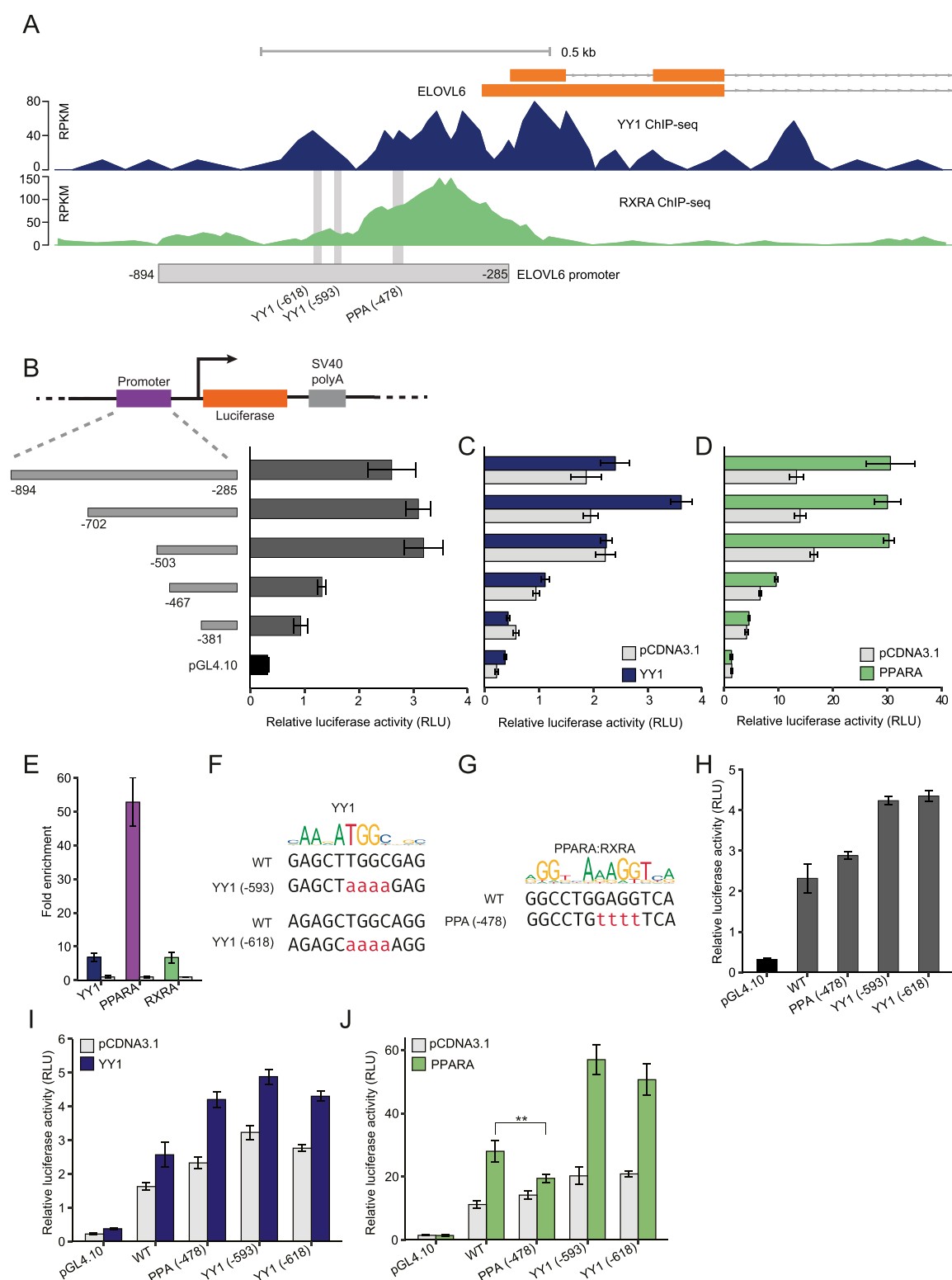

**Figure 9. Fine mapping and validation of the key genomic regions respond to Yin Yang 1 (*YY1*) and peroxisome proliferator activated receptor α (*PPARA*) overexpression in the *ELOVL6* promoter.**

**(A)** The YY1 and RXRA ChIP-seq signals in the promoter region of *ELOVL6* in HepG2 cells. The light grey box demonstrates the location of the promoter region (−894 to −285) examined in luciferase assay. The vertical light grey lines demonstrate the predicted binding sites for YY1 and PPARA/RXRA. **(B)** The minimal promoter region of *ELOVL6* is fine mapped to a ~200-bp region (luciferase construct −503 to −285 relative to the translational start site of *ELOVL6*) by luciferase assay with a series of truncation constructs. **(C, D)** The key regions of *ELOVL6* promoter respond to either *YY1* (C) or *PPARA* (D) overexpression were finely mapped by luciferase assay with the same set of

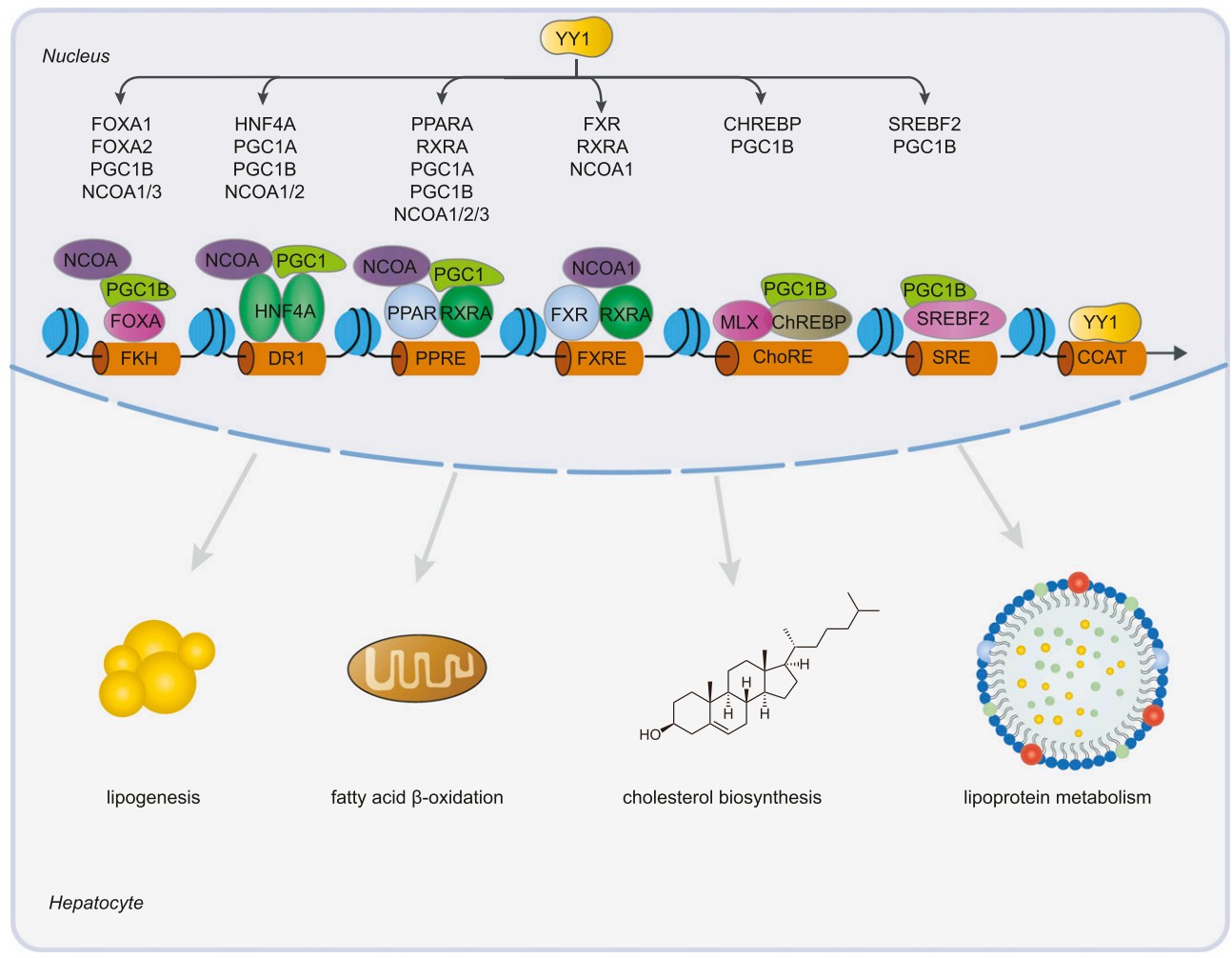

**Figure 10. Graphic representation of the regulatory relationships between Yin Yang 1 (YY1) and hepatic lipid metabolism.**
Knocking down of *YY1* down-regulated the expression of several key transcription factors and coactivators involved in hepatic lipid metabolism. Down-regulated transcription factors, including FOXA1/FOXA2, HNF4A, PPARA, RXRA, farnesoid X receptor (FXR), CHREBP, and SREBF2, decreased their binding to forkhead (FKH), direct repeat 1, PPAR response element (PPRE), FXR response element (FXRE), carbohydrate response element (ChoRE), and sterol regulatory element, respectively. This together with the decreased binding of YY1 to CCAT motif led to decreased expression of many key enzymes and proteins involved in lipid metabolism such as lipogenesis, fatty acid *β*-oxidation, cholesterol biosynthesis, and lipoprotein metabolism.

to be directly regulated by YY1 in our study. PPARA works as a metabolic sensor that switches its activity from coordination of lipogenesis in the fed state to promotion of FA uptake and FAO during the fasting state (Dubois et al, 2017). The down-regulation of *PPARA*, *RXRA*, and several of their coactivators including *PGC1A*, *PGC1B*, and *NCOA1/2/3* may have severely decreased the binding of the PPARA/RXRA complex to PPREs which further down-regulated the expression of nearby target genes (Fig 10). Previous studies have shown that PPARA regulates hepatic lipogenesis by either

activating the transcription of *SREBF1* or directly binding to the PPREs in the promoters of genes involved in lipogenesis, for example, *FADS2*, *MOD1*, and *SCD* (Miller & Ntambi, 1996; Tang et al, 2003; Rakhshandehroo et al, 2010; Pawlak et al, 2015). As the expression of *SREBF1* was determined unchanged by YY1 knockdown in our study, the observed down-regulation of genes in hepatic lipogenesis, genes in FA synthesis in particular, may be caused by the decreased binding of PPARA/RXRA complex directly to the PPREs in their promoters (Fig 4D and E). This is illustrated by the

truncation constructs. **(E)** ChIP-qPCR validation of YY1, PPARA and RXRA binding to the promoter region of *ELOVL6*. Fold changes of the target transcription factors (TFs) chromatin binding compared with IgG control are presented with GDCHR12 in Fig 6B and F used as the negative control. Error bars, SD; *n* = 3 technical replicates. **(F, G)** Detailed view of the predicted binding site for YY1 (F) and PPARA/RXRA (G) in the *ELOVL6* promoter. The detailed mutations introduced to the predicted TFs binding sties in luciferase constructs are highlighted in red. **(H)** Introducing mutations into the predicted TFs binding sites did not significantly decreased the promoter activity of *ELOVL6*. **(I, J)** The differences in the responses of the *ELOVL6* promoter to both *YY1* (I) and *PPARA* (J) overexpression after the mutations introduced. For (A, B, C, H, I, J), mean ± SD of six technical replicates from two independent plasmid extractions and transfections with each transfection had three technical replicates. **P < 0.01 calculated by two-tailed *t* tests.

identification of PPARA regulating *SCD* and *ELOVL6*, encoding key enzymes in monounsaturated FAs synthesis, by binding to functional PPREs in their promoters (Figs 8 and 9). However, the other isotypes of PPARs, including PPARD and PPARG, share the same heterodimeric RXR partners and the same coactivators of PPARA (Piccinin et al, 2019). They also bind to similar PPREs as the PPARA/RXRA complex does in the genome. Our current results cannot preclude the possibility that the down-regulation of genes in lipid metabolism by YY1 knockdown may be also caused by the decreased binding of the other PPAR/RXR complexes other than the PPARA/RXRA complex.

In addition to the previous observations in animal models, our data supported the hypothesis that the involvement of dysregulated *YY1* in NAFLD is mainly mediated by the dysregulation of the DNL pathway (Lu et al, 2014; Lai et al, 2018). Among the key genes in the DNL pathway, only *SCD* and *ELOVL6*, encoding key enzymes in monounsaturated FAs synthesis, were significantly down-regulated by YY1 knockdown (Fig 3B) (Strable & Ntambi, 2010). Increased expression of both *SCD* and *ELOVL6* activated the synthesis of monounsaturated FAs which promoted TG synthesis and contributed to the prevalence of NAFLD (Kotronen et al, 2009; Matsuzaka et al, 2012; Zhang et al, 2014). The detailed study on the cooperation between YY1 and PPARA/RXRA in determining the promoter activities of both the *SCD* and *ELOVL6* promoters illustrated that YY1 directly regulates key genes in the DNL which may further contribute to NAFLD progression. Overexpression of *YY1* was observed to inhibit the expression of *FXR* in obese mice, which further activated *SREBF1* expression and the DNL pathway (Lu et al, 2014). Similarly, stabilizing the expression of *YY1* was observed to significantly activate the DNL pathway through increased expression of *SREBF1* and *PPARG*, which further induced the early onset of NAFLD in zebrafish (Lai et al, 2018). In our study, the expression of *FXR* was significantly down-regulated by YY1 knockdown, whereas the expression of both *SREBF1* and *PPARG* were unchanged (Fig 4). Recently, YY1 has also been demonstrated to increase cellular TG and lipid accumulation in HCC cells through blocking the expression of *PGC1B*, which further suppressed FAO (Li et al, 2019). In contrast, our data showed that the expression of the key genes in mitochondria FAO were significantly down-regulated by YY1 knockdown with both *PGC1A* and *PGC1B* significantly down-regulated (Fig 4C and E). These seemingly contradictory results may be caused by differences in species, nutritional condition, and dosage of *YY1* expression. For example, the expression of *YY1* reduced to around one eighth by lentiviral-mediated knockdown in our study while the expression of *YY1* was around half by shRNA-mediated transient knockdown in the HCC study (Li et al, 2019). However, because all the previous studies focused on limited sets of genes and pathways, we currently cannot make detailed comparison between our study and previous studies.

In addition, we acknowledge that this study was carried out in an in-vitro cell line model that some aspects of the intrahepatocellular lipid metabolism may be different from in-vivo animal models and other cell lines (Gunn et al, 2017). For example, the well-known target genes of PPARA in mitochondrial FAO including *ACADVL*, *ACADM*, *ACADS*, *HADHA*, and *HADHB* were unchanged by YY1 knockdown in our RNA-seq, even though multiple other genes in mitochondrial FAO were down-regulated by the down-regulation of *PPARA* and its coactivators *PGC1A* and *PGC1B* (Fig 3A and D) (Rakhshandehroo et al,

2010). This may suggest that HepG2 cells are less suitable for charactering the metabolic changes in FAO. Further replicative studies in other hepatic cell lines or animal models are needed to clarify this inconsistency.

In conclusion, we identified a rigid set of DEGs induced by YY1 knockdown through high-throughput RNA-seq. We further unraveled the involvement of YY1 in regulating hepatic lipid metabolism by regulating the expression of several key TFs and their coactivators. The identified cooperation between YY1 and PPARA/RXRA complex in determining the *SCD* and *ELOVL6* promoter activities provide important mechanistic insights to the transcriptional regulation of DNL in the liver. Further studies are needed to elucidate the role of YY1 and its regulated TFs and coactivators in hepatic lipid metabolism and liver diseases, NAFLD and HCC in particular.

## Materials and Methods

### Cell culture

HepG2 cells were originally purchased from the American Type Culture Collection and maintained in RPMI1640 basal medium supplemented with 10% FBS and 2 mM L-glutamine. Human 293T cells were grown in DMEM supplemented with 10% FBS, 1 mM sodium pyruvate, and 500 μg/ml Geneticin. All the cells were also supplemented with 100 U of penicillin and 100 μg of streptomycin per 1 ml of culture medium.

### Lentiviral-mediated *YY1* knockdown

Both shRNA and amiRNA expression cassettes were used to knockdown *YY1* as previously described (Pan et al, 2020). The amiRNA precursor was designed to target GGGAGCAGAAGCAGGTGCAGAT and shRNA was designed to target GCCTCTCCTTTGTATATTATT of human *YY1* mRNA, respectively (Li et al, 2012). Lentivirus was produced in 293T cells by transfecting the lentiviral plasmid (pBMN-AS-YY1 or pBMN employed as control) together with packaging plasmids pLP1 and pLP2 and envelope plasmid pLP/VSVG (Life Technologies) using polyethylenimine (Polysciences) following the manufacturer's instructions. Cells were plated in 24-well plates and transduced with virus supernatant together with sequabrene (Sigma-Aldrich) at a final concentration of 8 μg/ml. The cells with stable YY1 knockdown were selected by Puromycin (Life Technologies) at a concentration of 1 μg/ml. The selected cells were maintained with 0.5 μg/ml of Puromycin and regularly passaged for further analysis.

### RNA-seq library preparation

Equal number of cells (~8 × $10^5$) were plated in 12-well plate 24-h before harvesting with TRIzol reagent (15596026; Thermo Fisher Scientific). Total RNA was isolated from TRIzol reagent with the PureLink RNA Mini Kit (12183018A; Thermo Fisher Scientific) following the manufacturer's instructions. The optional on-column PureLink DNase (12185010; Thermo Fisher Scientific) treatment was carried out to get rid of trace amount of genomic DNA during RNA extraction. The quality of the extracted total RNA was evaluated

using the Agilent 2200 TapeStation system. The library was constructed with QuantSeq 3′ mRNA-Seq Library Prep Kit REV for Illumina (016; Lexogen) following the protocol from the manufacture. The library was sequenced on an Illumina HiSeq 2500 sequencer (pair-end with 100 bp) in Macrogen. All the sequence data have been deposited in NCBI's Gene Expression Omnibus (GEO) and accessible through GEO Series accession number GSE158884.

### Processing of RNA-seq data

Quality of RNA-seq library was assessed using FastQC (0.11.9, https://www.bioinformatics.babraham.ac.uk/projects/fastqc/), adapter sequences and poly(A) tail sequences were removed using Trim Galore (0.6.4, http://www.bioinformatics.babraham.ac.uk/projects/trim_galore/). Reads were mapped to the human hg19 genome with *HISAT2* version 2.1.0 using –score-min L,0,-0.8 –no-unal parameters (Kim et al, 2019). Alignments were converted to sorted BAM files using SAMtools version 1.9 to keep only properly paired reads and also discard reads with mapping quality lower than 10 (Li et al, 2009). The gene-wise count matrix was generated by featureCounts version 1.6.4 with parameters –P –B –C –d 30 –D 800000 –s 2 (Liao et al, 2014). The R package DESeq2 v.1.26.0 was used for differential expression analysis (Love et al, 2014). The identified DE genes were subjected to enrichment analyses for GO terms, KEGG pathways, and GSEA by the R package clusterProfiler version 3.14.3 with highly similar GO terms filtered by the R package GOSemSim version 2.12.0 (Subramanian et al, 2005; Yu et al 2010, 2012).

### Reverse transcription and real-time quantitative PCR

We carried out RT-qPCR to validate the RNA-seq data. Two batches of samples, independent from the samples used in RNA-seq, were cultured in 12-well plate and harvested with TRIzol reagent (15596026; Thermo Fisher Scientific) followed by total RNA extraction with the PureLink RNA Mini Kit (12183018A; Thermo Fisher Scientific) as described in RNA-seq library preparation. A total of 1 $\mu$g total RNA was reverse transcribed into cDNA with Maxima First Strand cDNA synthesis kit (Thermo Fisher Scientific). The qPCR reactions were performed with JumpStart Taq ReadyMix (Sigma-Aldrich) coupled with EvaGreen dye (Biotium). The expression was normalized to two control transcripts *RSP18* and *ACTB*, respectively. Statistical analyses were carried out by two-tailed *t* tests. The detailed primers amplifying each target gene are listed in Table S4.

### YY1 ChIP-seq analyses

YY1 ChIP-seq experiments performed in liver tissue or liver-originated HepG2 cells were collected from the ENCODE project (The Encode Project Consortium, 2012). The data include YY1 ChIP-seq signal from the liver tissue of a 4-yr-old female (GEO: GSE96514), a 32-yr-old male (GEO: GSE96146) and HepG2 cells (GEO: GSM803381). The reads distribution of the ChIP-seq libraries over genes was visualized by *deeptools* version 3.1.3 (Ramírez et al, 2016). The genomic annotation of the identified YY1 peaks and intersection with the DE genes were carried out by the R package ChIPseeker version 1.22.1 (Yu et al, 2015). The binding of TFs enriched

over the DEGs with or without YY1 binding nearby were identified by the Enrichr web server (https://amp.pharm.mssm.edu/Enrichr/).

### tfNet analysis

We downloaded the full collection of TFs ChIP-seq and DNase-seq from ENCODE for HepG2 cells (Table S5). TFs represented by more than one experiment were combined by merging the overlapping TFs binding sites to avoid artifacts of overestimation of TFs co-occurrences. Next, we ran tfNet to detect a collection of putative regulatory regions and TF interactions in HepG2 cells (Diamanti et al, 2016). We clustered peaks located within 300 bp in the same regulatory regions and we considered only regions that harbored at least two peaks from different signals. TF peak summits located within 10 bp were considered for the overlapping TF interaction networks, whereas TF peak summits located 20–100 bp were considered for the neighboring TF interaction network. Bonferroni corrected *P*-values for the neighboring and overlapping TF pairs were calculated using the hypergeometric distribution.

### Overexpression of *PPARA* in 293T cells

To determine the in-vivo expression of PPARA, we overexpressed *PPARA* in human 293T cells as the positive control. Equal number of 293T cells (~8 × 10$^5$) were plated in six-well plate and transfected with 2 $\mu$g of pCDNA3.1-PPARA 24 h after plating using polyethylenimine (Polysciences) following the manufacturer's instructions. The cells were harvested 48 h posttransfection with RIPA buffer (R0278; Sigma-Aldrich). The nontreated human 293T cells were also harvested and used as the control in the following Western blot analysis.

### Western blot assays

Equal number of cells (~8 × 10$^5$) were plated in 12-well plate 48-h before harvesting with RIPA buffer (R0278; Sigma-Aldrich). The concentration of each protein extraction was quantified with Qubit protein assay kit (Q33212; Thermo Fisher Scientific). Western blotting analysis of protein extracts was performed as previously described (Pan et al, 2017). Around 20 µg protein extract was used for each target except for PPARA which used 60 µg protein extract. The primary antibodies used in this study are listed in Table S6. HRP-conjugated secondary antibodies (SC-2004, SC-2005, and SC-2354; Santa Cruz Biotechnology) were used to visualize the target protein under a CCD camera from Chemi-Doc XRS System (Bio-Rad).

### Chromatin immunoprecipitation

ChIP experiments were performed as previously described (Pan et al, 2020). Briefly, chromatin from 3 × 10$^6$ cells were incubated together with 8 µg corresponding antibodies to precipitate target regions. YY1 antibody SC-1703x (Santa Cruz Biotechnology), PPARA antibody SC-398394 (Santa Cruz Biotechnology), and RXRA antibody SC-553 (Santa Cruz Biotechnology) were used in this study. For each ChIP experiment, the same amount (8 µg) of normal rabbit IgG antibody (12-370; Milipore) was also incubated with sonicated chromatin from the same number of cells to check the background

of antibody nonspecific binding. Each experiment was replicated three times. The results are expressed as the relative enrichment of target DNA fragments immunoprecipitated by different TFs above fragment-specific background (IgG). The occupancy level of TFs normalized with IgG in the region GDCHR12 was used as the negative control as previous described (Pan et al, 2020). The detailed primers amplifying the target regions are listed in Table S7.

### Expression plasmids construction

The coding sequence of human *YY1* and *PPARA* were PCR amplified from cDNA reverse transcribed from HepG2 cells. Primers YY1-INF: GTTTAAACTTAAGCTTGCCgccatggcctcgggcgaca and YY1-INR: GCCACT-GTGCTGGATATCcttcccgtggtcgagaagggt were used to amplify the coding sequence of *YY1*. For *PPARA*, primers PPA-INF: TTAAACTT-AAGCTTGGTACCgtcgcgatggtggacacgga and PPA-INR: GCCACTGTGCTG-GATATCaaggaactcagtacatgtccct were used. The expression plasmids were constructed by inserting the amplicons into pCDNA3.1 through In-fusion cloning system (Takara). The resulting plasmids pCDNA3.1-YY1 and pCDNA3.1-PPARA were verified by Sanger sequencing.

To test the promoter activities, the putative promoter regions were inserted into pGL4.10 (Promega) digested with KpnI and EcoRV through In-fusion cloning system (Takara), except for the promoter region of *FADS1* which was constructed by T4 DNA ligation (New England Biolabs). Putative enhancer regions were inserted into pGL4.23 (Promega) digested with KpnI and EcoRV through In-fusion cloning system (Takara). A three fragments In-fusion cloning system with mutations introduced through primer sequences was used to introduce desired mutations into luciferase constructs. The detailed primer information and cloning method for each construct is listed in Table S8. All the resulting plasmids were verified by Sanger sequencing.

### Luciferase reporter assay

Luciferase plasmids were purified with GenElute Plasmid Miniprep Kit (Sigma-Aldrich) and quantified by NanoDrop 2000 (Thermo Fisher Scientific). HepG2 cells were plated 1 d before transfection in 96-well plates. The confluency was 70% on transfection. For normal luciferase assay, each well was transfected with 0.3 $\mu$l XtremeGENE HP DNA transfection reagent (Roche) and 100 ng of experimental firefly luciferase reporter plasmid, and 1 ng of pGL4.74 renilla luciferase reporter vector (Promega) as internal control for monitoring transfection and lysis efficiency. For luciferase assay overexpressing YY1, each well was transfected with 0.3 $\mu$l Xtre-meGENE HP DNA transfection reagent, 50 ng of firefly luciferase reporter plasmid, 50 ng pCDNA3.1-YY1, and 1 ng of pGL4.74. For luciferase assay overexpressing PPARA, each well was transfected with 0.3 $\mu$l XtremeGENE HP DNA transfection reagent, 40 ng of firefly luciferase reporter plasmid, 50 ng pCDNA3.1-PPARA, and 10 ng of pRL-MP described earlier (Pan et al, 2017). For experiments cotransfecting YY1 and PPARA, each well was transfected with 25 ng of each expression plasmid together with 50 ng luciferase plasmid and 1 ng pGL4.74. Plasmid pCDNA3.1 was used as the control expression plasmid when needed.

Cells were harvested 24 h after transfection and assayed with the Dual-Luciferase Reporter Assay System (Promega) on an Infinite M200 pro reader (Tecan). All the results are expressed directly as the ratio of firefly luciferase activity from experiment plasmids to renilla luciferase activity from control plasmids. All the luciferase experiments came from two independent transfections, that is, independent plasmid preparations and transfections each with three technical replicates. Luciferase values are expressed as averages with error bars representing SDs from all technical replicates and statistical analyses were performed by two-tailed *t* tests.

## Data Availability

Transcriptome data from this study are available at GEO under accession number GSE158884. The lentiviral plasmid pBMN-AS-YY1 used for *YY1* knockdown in this study is deposited to Addgene (plasmid # 154943). The expression plasmids pCDNA3.1-YY1 and pCDNA3.1-PPARA have also been deposited to Addgene and assigned the identifier number 169018 and 169019, respectively.

## Supplementary Information

## Acknowledgements

This study was supported by grants from Swedish Cancer Foundation (CAN 2018/849), the Swedish Diabetes Foundation (DIA 2017-269), and the Family Ernfors Fund and EXODIAB to C Wadelius. J Komorowski and K Diamanti were partially supported by a grant from the eSSence consortium and by the Institute of Computer Science, Polish Academy of Sciences.

### Author Contributions

G Pan: conceptualization, formal analysis, investigation, visualization, methodology, and writing—original draft, review, and editing.
K Diamanti: formal analysis and writing—review and editing.
M Cavalli: visualization and writing—review and editing.
A Lara Gutiérrez: investigation.
J Komorowski: visualization and writing—review and editing.
C Wadelius: conceptualization, supervision, funding acquisition, methodology, project administration, and writing—review and editing.

### Conflict of Interest Statement

The authors declare that they have no conflict of interest.

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
