## [Reviewer comments · Life Science Alliance]

Life Science Alliance

Multifaceted regulation of hepatic lipid metabolism by YY1

Gang Pan, Klev Diamanti, Marco Cavalli, Ariadna Lara Gutierrez, Jan Komorowski, and Claes Wadelius

DOI: <https://doi.org/10.26508/lsa.202000928>

Corresponding author(s): Claes Wadelius, Science for Life Laboratory, Department of Immunology, Genetics and Pathology, Uppsala University, Uppsala, Sweden and Gang Pan, Uppsala universitet

Review Timeline:

Submission Date:	2020-10-05
Editorial Decision:	2021-01-14
Revision Received:	2021-04-14
Editorial Decision:	2021-05-20
Revision Received:	2021-05-25
Accepted:	2021-05-25

Scientific Editor: Shachi Bhatt

Transaction Report:

January 14, 2021

Re: Life Science Alliance manuscript #LSA-2020-00928

Prof. Claes Wadelius
Science for Life Laboratory, Department of Immunology, Genetics and Pathology, Uppsala
University, Uppsala, Sweden
BMC Husargatan 3
Uppsala 75122
Sweden

Dear Dr. Wadelius,

Thank you for submitting your manuscript entitled "YY1 regulates hepatic De novo lipogenesis by cooperating with PPARA" to Life Science Alliance (LSA). The manuscript was assessed by expert reviewers, whose comments are appended to this letter.

We apologize for this extended delay in getting back to you. As you will note from the reviewers' comments below, the reviewers see value in your study, but have also raised a number of concerns that must be fully addressed prior to further consideration of the manuscript at LSA. Particularly, the identity of the band on westerns as PPAR α needs to be definitively substantiated, all other comments should be addressed too. We would also encourage you to not use acronyms in the abstract unless they are defined. We would thus like to invite you to submit a revised version of the manuscript that addresses all of the reviewers' concerns.

Thank you for this interesting contribution to Life Science Alliance. We are looking forward to receiving your revised manuscript.

Sincerely,

Shachi Bhatt, Ph.D.
Executive Editor
Life Science Alliance
<https://www.lsjournal.org/>
Tweet @SciBhatt @LSAJournal

- A letter addressing the reviewers' comments point by point.
- An editable version of the final text (.DOC or .DOCX) is needed for copyediting (no PDFs).
- High-resolution figure, supplementary figure and video files uploaded as individual files: See our detailed guidelines for preparing your production-ready images, <https://www.life-science-alliance.org/authors>
- Summary blurb (enter in submission system): A short text summarizing in a single sentence the study (max. 200 characters including spaces). This text is used in conjunction with the titles of papers, hence should be informative and complementary to the title and running title. It should describe the context and significance of the findings for a general readership; it should be written in the present tense and refer to the work in the third person. Author names should not be mentioned.

B. MANUSCRIPT ORGANIZATION AND FORMATTING:

Reviewer #1 (Comments to the Authors (Required)):

Ying Yang 1 (YY1) has been implicated in liver pathophysiology. Increased YY1 expression is correlated with NAFLD and liver cancer, potentially contributing to the enhanced de novo lipogenesis (DNL) and attenuated fatty acid oxidation (FAO). In this study, the authors generated RNAseq dataset of YY1-knocked down HepG2 cell line, which is characterized by alteration of fatty

acid metabolism and PPAR signaling pathway genes. It was also demonstrated that PPARalpha expression was regulated by YY1, which cooperatively acts with YY1 to modulate fatty acid metabolism genes. It is of interests that this study suggested a molecular mechanism of YY1 on liver pathology. But, at the same time, several points significantly weakened the central idea of YY1 action on PPARalpha function.

1. Given the bona fide function of PPARalpha on hepatic FAO, these results cannot fully support the idea on the reduction of fatty acid accumulation. In the initial data (Fig 1-3), PPAR signaling pathway has been impacted by YY1 knock-down. However, latter focus on PPARalpha may need more justification. The authors suggested PPARalpha expression was solely decreased in YY1 knock-down but it is not sufficient (see #5). Therefore, the major claim of this study on the PPARalpha regulation of DNL and fatty acid accumulation may be a significant weakness.
2. As discussed, few reports suggested PPARalpha increases some lipogenic gene expressions. However, this lipogenic function has been explained to only serve to counteract the excessive fatty acid breakdown and/or to change fatty acid composition (PMID14999402). It is more likely that PPARalpha inactivation increases overall fat accumulation. Especially, it is not aligned with the therapeutic usage of PPARalpha ligands (e.g., fibrates) against NAFLD.
3. Related to #2, the most well-known PPARalpha targets such as Acox1, Acot1, Acot3, and Fgf21 should be compared, to validate the changes of PPARalpha activity. Are potential new targets in DNL pathway (Fig 3B and 5B) regulated by direct PPARalpha activation by ligands? GW7647, Wy-14643, and/or fibrates.
4. It looks like that blunted PGC1 and/or Rxra expressions (Fig S1 and 4B) might be associated with the reduction of broad PPAR signaling pathway (both alpha/gamma) rather than specific inactivation of PPARalpha. Do the authors have any considerations on the potential role of PGC1 on overall PPAR function, although contradicted (PMID: 31695789)?
5. Like other nuclear receptors, transcriptional activity of PPARalpha (or others) is largely dependent on the ligand availability rather than its expression. Some key experiments should be repeated with PPARalpha ligands. To this end, although the authors ruled out some key lipogenic transcription factors (like LXR), it should be re-considered that ligand availability under specific condition might contribute to the overall reduction of lipogenic pathway in YY1 knock-down cells.

Reviewer #2 (Comments to the Authors (Required)):

The paper has examined the effect of YY1 knockdown in HepG2 cells, using RNA sequencing as a starting point. The detailed studies reveal an interaction between YY1 and PPARa in the regulation of lipogenic gene expression. The paper is well written and experimentally thorough, with some minor exceptions. The paper represents a valuable addition to the literature. Nevertheless, a number of concerns can be raised.

Major comments

1) PPARa is notoriously difficult to detect by Western blot. Figure 4C present the results of Western blots with two different antibodies. How can the authors be sure that the band at 70 kD is indeed PPARa. It should actually migrate at around 55 kD. Also, all the bands in the Western blot with the Santa Cruz antibody are reduced upon YY1 knockdown. Proper controls should be included, for instance by overexpressing PPARa in HepG2 cells. This should be comparatively easy, as the authors have done overexpression studies in figure 5.

2) PPARa is believed to protect against NAFLD. In line with that notion, drugs targeting PPARa are

being investigated as a treatment for NAFLD. How does the regulation of PPAR α by YY1 fit into this picture and the purported stimulatory effect of PPAR α on lipogenesis?

Related to this, the paper focuses on the regulation of lipogenic genes by YY1 via PPAR α . What about genes involved in fatty acid oxidation and ketogenesis? These genes are activated by PPAR α but seem to be repressed by YY1. To what extent is the interaction between YY1 and PPAR α expected to be generic (thus for all PPAR α target genes) or limited to specific promoters and through which mechanism? These issues should be addressed in the discussion (at the expense of basic info on PPARs)

3) One of the limitations of the paper is the use of HepG2 cells, which are well recognized to be a relatively poor model for studying PPAR α -mediated gene regulation. For example, HepaRG cells much better capture the extensive role of PPAR α in the regulation of hepatic lipid metabolism. The authors should mention this limitation in the discussion and elaborate on to what extent the use of HepG2 may explain why key genes in mitochondrial β -oxidation were unchanged by YY1 knock-down. It is well known that the induction of fatty acid oxidation by PPAR α is not very well reflected in HepG2 cells.

Minor comments

1) Figure 3B. It is incorrect to classify ANGPTL4 as adipocyte differentiation. A better description would be " plasma triglyceride metabolism"

2) The paper is quite wordy, in particular the results and discussion. There is no need to elaborate on the basic of PPARs in the discussion (line 413-418). Authors are encouraged to shorten the results and discussion sections.

3) Line 77, Please note that chylomicron remnants carry dietary fatty acids to the liver.

4) Line 120, please change principle to principal.

5) Line 170-173, please rephrase or split in two.

6) Line 177-179, please rephrase

7) Line 703, reference is incomplete.

Authors' response to the review comments

We greatly appreciate the time and efforts by the editor and referees in reviewing this manuscript. We agree with the comments from both of the reviewers and have made necessary experiments and/or added explaining text to meet the criticism. In addition to the changes mentioned in the point to point response below, we want to emphasize that to facilitate the review process, we have kept the track changes in our revised manuscript. The line number mentions below denotes the lines showed under "simple markup" mode in the Word Review panel.

Reviewer #1:

Ying Yang 1 (YY1) has been implicated in liver pathophysiology. Increased YY1 expression is correlated with NAFLD and liver cancer, potentially contributing to the enhanced de novo lipogenesis (DNL) and attenuated fatty acid oxidation (FAO). In this study, the authors generated RNAseq dataset of YY1-knocked down HepG2 cell line, which is characterized by alteration of fatty acid metabolism and PPAR signaling pathway genes. It was also demonstrated that PPARalpha expression was regulated by YY1, which cooperatively acts with YY1 to modulate fatty acid metabolism genes. It is of interests that this study suggested a molecular mechanism of YY1 on liver pathology. But, at the same time, several points significantly weakened the central idea of YY1 action on PPARalpha function.

Response:

We thank the reviewer for finding our manuscript of interest to the understanding of dysregulated YY1 in various liver diseases. We also thank the reviewer for taking the time to review our manuscript.

1. Given the bona fide function of PPARalpha on hepatic FAO, these results cannot fully support the idea on the reduction of fatty acid accumulation. In the initial data (Fig 1-3), PPAR signaling pathway has been impacted by YY1 knock-down. However, latter focus on PPARalpha may need more justification. The authors suggested PPARalpha expression was solely decreased in YY1 knock-down but it is not sufficient (see #5). Therefore, the major claim of this study on the PPARalpha regulation of DNL and fatty acid accumulation may be a significant weakness.

Response:

We thank the reviewer for this insightful comment. We agree with the reviewer's opinion regarding the validity of focusing on PPARA only in following study. To get a deeper understanding of the molecular links between YY1 knockdown and dysregulated lipid metabolism in our RNA-seq data (Fig 2), we systematically evaluated the expression changes of all the genes in lipid metabolism based on annotations from the Reactome

database (Jassal et al 2020). To solve the conflicts between downregulated PPARA and lacking of differentially expressed genes (DEGs) in FAO, we further relaxed our statistical threshold in determining DEGs in RNA-seq from $\log_2\text{foldchange} > 1$ and P value < 0.001 to $\log_2\text{foldchange} > 0.7$ and P value < 0.001 . Our analysis indeed found multiple genes in mitochondrial and peroxisomal FAO downregulated by YY1 knockdown, the downregulation of which were further validated by RT-qPCR in two batches of samples (n = 8) independent from the samples used in RNA-seq (Fig 3). Even though our new results confirmed that multiple genes encoding important enzymes in FAO e.g. *ECH1*, *HADH*, *ACAA2*, *ACADL* and *ACADSB* were downregulated by YY1 knockdown, we acknowledged that several of the well-known target genes regulated by PPARA including *ACADVL*, *ACADM*, *ACADS*, *HADHA* and *HADHB*, were detected unchanged in our RNA-seq (Rakhshandehroo et al 2010). As pointed out by reviewer 2, these discrepancies might be caused by the fact that HepG2 cells are less suitable as a model in characterizing the metabolic changes in FAO. To address this possible shortcoming, we have added a short paragraph in the discussion section on this topic (line 507-515).

In addition to FAO, our new results confirmed that there are multiple other biological processes in lipid metabolism including fatty acid synthesis (FAS) and transport, cholesterol biosynthesis and lipoprotein metabolism downregulated by YY1 knockdown (Fig 3). Further investigation on the dysregulated transcription factors (TFs) involved in regulating lipid metabolism revealed that multiple TFs including the RXRA and its heterodimeric partners PPARA and FXR, CHREBP, SREBF2, HNF4A and FOXA1/2 were all downregulated by YY1 knockdown (Fig 4). The downregulation of these TFs were validated by both RT-qPCR and Western blot analysis (Fig 4). Based on the new findings, we have accordingly changed our hypothesis regarding how YY1 regulating lipid metabolism in our revised manuscript.

2. As discussed, few reports suggested PPARalpha increases some lipogenic gene expressions. However, this lipogenic function has been explained to only serve to counteract the excessive fatty acid breakdown and/or to change fatty acid composition (PMID14999402). It is more likely that PPARalpha inactivation increases overall fat accumulation. Especially, it is not aligned with the therapeutic usage of PPARalpha ligands (e.g., fibrates) against NAFLD.

Response:

We thank the reviewer for raising this question. To answer the reviewer's concern, we first thoroughly examined the expression changes of all genes in the DNL pathway. We confirmed that knocking down of YY1 significantly downregulated the expression of multiple genes in the FAS pathway, the expression of which were determined by both RT-qPCR and Western blot analysis (Figs 3B and H). The genes encoding enzymes in glucose metabolism that generate acetyl-CoA as the starting point of DNL and encoding enzymes in triglycerides (TG) synthesis were generally unchanged by YY1 knockdown. Our previous claim that YY1 regulates the DNL pathway was inaccurate and was amended accordingly in our revised manuscript.

We agree that the function of PPARA was most well characterized in FAO and lipoprotein metabolism. The role of PPARA in lipogenesis is less clear so far even

though there is mounting evidence suggests that PPARA is also involved in hepatic lipogenesis process. In the fed state, PPARA has been reported to coordinate different pathways of DNL to supply FA for storage as hepatic TG, for periods of starvation; while during fasting, PPARA shifts its activity to promote FA uptake and oxidation, thus yielding substrate for ketone body synthesis to provide energy for peripheral tissues (Pawlak et al 2015). SREBP1, coded by *SREBF1*, works as one of the master regulators in hepatic lipogenesis pathways. PPARA agonists were reported to increase human *SREBF1* transcription through increasing PPARA binding directly to its promoter region (Fernandez-Alvarez et al 2011). In addition to direct regulation, PPARA was also observed to enhance the activity of SREBP1 through multiple other mechanisms (Dubois et al 2017, Pawlak et al 2015). Accordingly, multiple genes in the lipogenesis pathways were reported to be directly or indirectly regulated by PPARA (Rakhshandehroo et al 2010).

By examining the expression changes of genes involved in hepatic lipogenesis, we observed that the DEGs induced by YY1 knockdown were enriched in FAS pathways (Figs 3A, B and H). Even though the role of PPARA in hepatic lipogenesis is less clear, its involvement in regulating the FAS pathways were supported by multiple studies (Burri et al 2010, Dubois et al 2017, Pawlak et al 2015, Rakhshandehroo et al 2009). As many of the unsaturated FAs are important PPARA agonists, the activation of the FAS pathways by PPARA was suggested to ensure enough FAs agonists for its own activation (Burri et al 2010). Our results on the characterization of the cooperation between YY1 and PPARA/RXRA in regulating the promoter activities of the human *SCD* and *ELOVL6* promoters confirmed the previous observed dual regulation by SREBP1 and PPARA for the genes involved in unsaturated fatty acids synthesis (Mandard et al 2004, Pawlak et al 2015). Additionally, the fine mapping of the regulatory elements in the human *SCD* promoter also replicated the previous findings in the mouse that PPARA regulates *SCD* expression by directly binding to the PPRES in the *SCD* promoter region (Miller & Ntambi 1996). To clarify our aim and purpose in studying the cooperation between YY1 and PPARA, we made a detailed explanation regarding this topic in line 310-321 in our manuscript.

It should be noted that even though the expression of the other PPARs were not changed by YY1 knockdown, the downregulation of RXRA and several of their transcriptional coactivators including PGC1A, PGC1B and NCOA1/2/3 may still affect the regulatory activities of the other PPARs. Our current results can't exclude the possibility that the downregulation of the identified PPARA target genes may also be regulated by other PPARs. To address these potential interferences, we have added a short discussion in line 476-481 regarding this.

3. Related to #2, the most well-known PPARalpha targets such as *Acox1*, *Acot1*, *Acot3*, and *Fgf21* should be compared, to validate the changes of PPARalpha activity. Are potential new targets in DNL pathway (Fig 3B and 5B) regulated by direct PPARalpha activation by ligands? GW7647, Wy-14643, and/or fibrates.

Response:

We thank the reviewer for bringing up this point. By exploring our RNA-seq results, we found that two of the reviewer mentioned PPARA target genes i.e. *ACOT3* and *FGF21*

were not detected in our RNA-seq analysis. The other two genes were determined unchanged by YY1 knockdown with padj value of 0.03 for ACOX1 and padj value of 0.19 for ACOT1, which is also expressed at very low level with base mean of 2.2 in our RNA-seq. PPARA regulates the expression of human liver fatty acid binding protein (*FABP1*) by directly binding to the PPRES in its promoter region (Guzmán et al 2013). As a gene directly regulated by PPARA, we have chosen *FABP1* as the positive control in studying PPARA regulation in our study (Figs 6E-I). The results of ChIP-qPCR with antibody against PPARA and luciferase assay on human *FABP1* promoter responding to PPARA overexpression replicated the previous findings and worked as the positive control in charactering the promoter regions of both *SCD* and *ELOVL6* (Figs 8 and 9) (Guzmán et al 2013).

Among the investigated seven genes encoding key enzymes in FAS pathways (previous Fig 5B and now in Fig 3B), six of them (except *ELOVL2*) have already been verified as PPARA target genes in multiple studies carried out in human, mouse and rat mainly with PPARA agonist Wy14643 (Rakhshandehroo et al 2010). These observations together with the enriched binding of RXRA nearby these genes are the main reasons we choose to study whether they are directly regulated by PPARA. For other DEGs in the PPAR signaling pathway (Fig 3B in our initial manuscript), all of them have also been validated as PPARA targets in multiple studies that are carried out in human or mouse or both (Rakhshandehroo et al 2010).

4. It looks like that blunted PGC1 and/or Rxra expressions (Fig S1 and 4B) might be associated with the reduction of broad PPAR signaling pathway (both alpha/gamma) rather than specific inactivation of PPARalpha. Do the authors have any considerations on the potential role of PGC1 on overall PPAR function, although contradicted (PMID: 31695789)?

Response:

We agree with the reviewer's comment on this point. This is an important question which makes us realize that our previous conclusion was questionable. Even though the expression of other nuclear receptors e.g., PPARG was not affected by YY1 knockdown, the downregulation of their heterodimeric partner RXRA may still significantly change their regulatory activity (Figs 4B and E). Additionally, the downregulation of nuclear receptor coactivators, i.e. PGC1A and PGC1B together with NCOA1/2/3, may further decrease the regulatory activities of these nuclear receptors (Figs 4C and E). The molecular function of PGC1A and PGC1B has been studied in detail. PGC1A and PGC1B regulate overlapping and distinct hepatic metabolism-related genes by directly docking on many different TFs to regulate their transcriptional activities (Lin et al 2005, Piccinin et al 2019). These two coactivators are directly involved in different aspects of liver metabolism, such as FAO, gluconeogenesis, mitochondrial biogenesis, and FAS. For example, both PGC1A and PGC1B are essential coactivators in activating FAO process mediated by PPARA. PGC1B promotes genes in DNL in the liver by directly co-activating SREBP1 and LXRA. Dysregulation of both PGC1A and PGC1B has been linked to the initiation and/or progression of liver steatosis, nonalcoholic fatty liver disease (NAFLD), nonalcoholic steatohepatitis (NASH) and

hepatocellular carcinoma (HCC) (Piccinin et al 2019). Based on the abovementioned observations, we conclude that the downregulation of several important TFs and their coactivators together with the downregulation of YY1 led to the downregulation of various genes in lipid metabolism (Figs 3 and 4). This hypothesis is supported by the Enrichr analysis of TFs downregulated by YY1 knockdown enriched nearby the downregulated genes, especially the 137 DEGs in lipid metabolism (Figs 5G and I). Through exploring the literature, we did find several studies contradicting with our findings including the reviewer mentioned PMID: 31695789 and we have add a paragraph in line 491-506 to discuss the possible causes of these discrepancies.

5. Like other nuclear receptors, transcriptional activity of PPARalpha (or others) is largely dependent on the ligand availability rather than its expression. Some key experiments should be repeated with PPARalpha ligands. To this end, although the authors ruled out some key lipogenic transcription factors (like LXR), it should be re-considered that ligand availability under specific condition might contribute to the overall reduction of lipogenic pathway in YY1 knock-down cells.

Response:

This is an interesting question raised by the reviewer. As mentioned in the response to point 2, many of the unsaturated FAs that transported by FABP1 are important natural PPARA ligands for its own activation (Burri et al 2010). The downregulation of genes in FAS pathways will decreased the amount of unsaturated FAs synthesized, which may further decrease the activity of PPARA. Additionally, the PGC1A and PGC1B coactivators are also essential for PPARA activation, the downregulation of these two coactivators may further decrease its regulatory activities.

The target genes regulated by PPARA in the liver have been well studied in both human and mouse models in many studies (Rakhshandehroo et al 2010). Beside from PPARA, several other important TFs and their coactivators were identified as downregulated by YY1 knockdown (Fig 4). Our current results support the hypothesis that YY1 directly or indirectly regulate the expression of several important TFs and their coactivators in hepatic lipid metabolism (Fig 10). Dysregulation of YY1 observed in many liver diseases may be caused by the dysregulation of several TFs and their coactivators regulated by YY1, which further affect hepatic lipid metabolism by dysregulating target genes in cooperation with YY1. Due to these reasons, we think that a systematic investigation of PPARA target genes may be less relevant in current phase.

Reviewer #2:

The paper has examined the effect of YY1 knockdown in HepG2 cells, using RNA sequencing as a starting point. The detailed studies reveal an interaction between YY1 and PPARa in the regulation of lipogenic gene expression. The paper is well written and experimentally thorough, with some minor exceptions. The paper represents a valuable addition to the literature. Nevertheless, a number of concerns can be raised.

Response:

We thank the reviewer for appreciating the robustness of our experimental strategy, and for taking the time to review our manuscript in such a thorough manner.

Major comments

1) PPAR α is notoriously difficult to detect by Western blot. Figure 4C present the results of Western blots with two different antibodies. How can the authors be sure that the band at 70 kD is indeed PPAR α . It should actually migrate at around 55 kD. Also, all the bands in the Western blot with the Santa Cruz antibody are reduced upon YY1 knockdown. Proper controls should be included, for instance by overexpressing PPAR α in HepG2 cells. This should be comparatively easy, as the authors have done overexpression studies in figure 5.

Response:

We truly appreciate that the reviewer pointing this out. Without further digging into the literature, we were misled by the datasheets of the antibodies from the manufactures. To verify the correct size of PPAR α in Western blot, we overexpressed PPAR α in human 293T cells. The expression of PPAR α was determined by Western blot with two different antibodies sc398394 and sc1985 from Santa Cruz Biotechnology. Western blot analysis in 293T cells showed that overexpression of PPAR α significantly increased its protein expression as the native 52-kDa form (Fig 4F) that is in accordance with previous observation (Passilly et al 1999). Knocking down of YY1 specifically decreased the activated 59-kDa form of PPAR α but not the 52-kDa form (Fig 4E). As the 59-kDa form of PPAR α is activated by Wy-14643, this finding suggested that the regulatory activity of PPAR α in YY1-knockdown cells is compromised (Passilly et al 1999). To help other people interested in replicating our results, we have deposited the pCDNA3.1-PPAR α plasmid to Addgene under identification number 169019. For the antibody PA1-822A from Thermo Fisher Scientific that is employed to detect PPAR α in our initial manuscript, it failed to detect the overexpressed PPAR α in 293T cells in our Western blot analysis which was discarded in our revision manuscript. Additionally, we have carefully checked all the other new Western blot results and found them in accordance with the literatures.

2) PPAR α is believed to protect against NAFLD. In line with that notion, drugs targeting PPAR α are being investigated as a treatment for NAFLD. How does the regulation of PPAR α by YY1 fit into this picture and the purported stimulatory effect of PPAR α on lipogenesis?

Related to this, the paper focuses on the regulation of lipogenic genes by YY1 via PPAR α . What about genes involved in fatty acid oxidation and ketogenesis? These genes are activated by PPAR α but seem to be repressed by YY1. To what extent is the interaction between YY1 and PPAR α expected to be generic (thus for all PPAR α target genes) or limited to specific promoters and through which mechanism? These issues should be addressed in the discussion (at the expense of basic info on PPARs)

Response:

We thank the reviewer for these insightful comments. After carefully considering the comments from both reviewers and exploring the literatures, we agree that our current results can't totally support the previously claimed YY1-PPARA-SCD/ELOVL6 regulatory axis in lipogenesis. Without going into the details of dysregulation caused by YY1 knockdown, our previous hypothesis is based on the observation that the PPAR signaling pathway (KEGG: hsa03320) was the most significantly downregulated pathway and among the three isoforms of PPAR, only PPARA was downregulated by YY1 knockdown. We agree that the most well-known regulatory targets of PPARA are genes in FAO and lipoprotein metabolism. To solve the conflicts between downregulation of PPARA and lacking of downregulated genes in FAO, we relaxed our statistical threshold in determining DEGs in RNA-seq from $\log_2\text{foldchange} > 1$ and $P \text{ value} < 0.001$ to $\log_2\text{foldchange} > 0.7$ and $P \text{ value} < 0.001$. Our new results revealed that there are indeed multiple genes in FAO downregulated by YY1 knockdown which were further validated by RT-qPCR (Figs 3A and 3D). Many of the downregulated genes including *ACAA2*, *CPT1A*, *DECR1*, *ACAT1*, *ACOT2*, *HADH* and *ACADL* were known to be directly regulated by PPARA (Burri et al 2010, Rakhshandehroo et al 2010). We noted that even though the expression of the other isoforms of PPAR including PPARG and PPARD were not downregulated by YY1 knockdown, their downstream target genes may still be downregulated by the downregulation of their heterodimeric partner RXRA and their coactivators PGC1A and PGC1B. As the PPAR and RXR complex recognize the same PPREs, our current results can't exclude the possibility that the downregulation of the identified PPARA target genes may also be regulated by other PPARs. However, the identification of YY1 cooperating with PPARA and RXRA complex in regulating *SCD* and *ELOVL6* expression over their promoter regions are still valid (Figs 8 and 9). Multiple recent studies clearly proved that PPARA is directly regulating several key enzymes in FA synthesis e.g. *FADS2*, *SCD* and *MOD1* by binding to the PPREs in their promoters (Burri et al 2010, Dubois et al 2017, Pawlak et al 2015, Rakhshandehroo et al 2009). The seemingly paradox of PPARA regulating both FAO and FA synthesis are believed to be caused by the fact that the final products of the FA synthesis pathway are important PPARA agonists, the activation of the FA synthesis pathways by PPARA was suggested to ensure enough FAs agonists for its own activation (Burri et al 2010). The cooperation between YY1 and PPARA/RXRA complex in regulating the promoter activities of both *SCD* and *ELOVL6* promoters confirmed the previously identified regulation of *SCD* by PPARA binding to its promoter region in mouse (Miller & Ntambi, 1996). Our RNA-seq results with relaxed threshold also proved that multiple TFs and their coactivators that are important in hepatic lipid metabolism were dysregulated by YY1 knockdown (Fig 4). We believed that the involvement of YY1 in NAFLD and other liver diseases lays with these dysregulated TFs and coactivators. YY1 may directly or indirectly regulate the expression of several important TFs in lipogenesis which may further contribute to the prevalence of NAFLD. For example, the downregulation of RXRA and PGC1B may compromise the expression of genes in lipogenesis regulated by PPARs, especially PPARG. CHREBP, as one of the master regulator in lipogenesis, was downregulated by YY1 knockdown (Figure 4D and E). The expression of SREBF1, encoding the other master regulator of lipogenesis, was unchanged by YY1 knockdown. However due to the downregulation of its coactivator PGC1B, the SREBP1 target genes may be still downregulated by YY1 knockdown (Piccinin et al 2019). Even though

SREBP2 is regarded as the master regulator in cholesterol biosynthesis, newly emerging results indicated that SREBP1 and SREBP2 may have larger functional overlapping than previously thought (Piccinin et al 2019). As they recognize similar SREs, the downregulation of SREBF2 may also affect the expression of genes regulated by SREBP1. Lastly, downregulation of HNF4A may also contributed to the downregulation of genes in lipogenesis (Yin et al 2011).

In conclusion, our new results confirmed that the cooperation between YY1 and PPARA/RXR are important in regulating the genes in FA synthesis. The enriched binding of PPARA and RXR nearby genes downregulated by YY1 knockdown, especially the genes involved in hepatic lipid metabolism (Figs 5G and I) suggested that the downregulation of PPARA, RXRA and its coactivators PGC1A, PGC1B and NCOA1/2/3 may have a large effect on the downregulation of nearby target genes. Based on these new findings, we have accordingly changed our conclusions regarding the regulatory axis in our revision.

3) One the limitations of the paper is the use of HepG2 cells, which are well recognized to be a relatively poor model for studying PPARa-mediated gene regulation. For example, HepaRG cells much better capture the extensive role of PPARa in the regulation of hepatic lipid metabolism. The authors should mention this limitation in the discussion and elaborate on to what extent the use of HepG2 may explain why key genes in mitochondrial β -oxidation were unchanged by YY1 knock-down. It is well known that the induction of fatty acid oxidation by PPARa is not very well reflected in HepG2 cells.

Response:

We thank the reviewer for raising this question. We agree with the reviewer's opinion regarding the shortcoming of using HepG2 cells in studying FAO. Compared with other biological processes in lipid metabolism e.g., fatty acid synthesis, cholesterol biosynthesis and lipoprotein metabolism, the mitochondrial FAO seemed to be less affected by YY1 knockdown. By employing a less stringent threshold, we did find that multiple genes encoding important enzymes in mitochondrial FAO were significantly downregulated by YY1 knockdown in RNA-seq (Figure 3A). The downregulated genes include *ACADL*, *ACADSB*, *ECHS1*, *HADH* and *ACAA2* which encoding important enzymes in the initial steps of mitochondrial FAO (Houten et al 2016). However, we also noticed that the expression of *ACADS*, *ACADM*, *ACADVL*, *HADHA* and *HADHB*, also encoding key enzymes in the initial steps of mitochondrial FAO, were detected unchanged. These genes are all well-known targets that are directly regulated by PPARA (Rakhshandehroo et al 2010). To address these inconsistencies or shortcomings, we have added a short paragraph in the Discussion section from line 505 to 512 as follows:

"In addition, we acknowledge that this study was carried out in an in-vitro cell line model that some aspects of the intrahepatocellular lipid metabolism may be different from in-vivo animal models and other cell lines (Gunn et al 2017). For example, the well-known target genes of PPARA in mitochondrial FAO including *ACADVL*, *ACADM*, *ACADS*, *HADHA* and *HADHB* were detected unchanged by YY1 knockdown in our RNA-seq, even though multiple other genes in mitochondrial FAO were downregulated by the

downregulation of PPARA and its coactivators PGC1A and PGC1B (Figure 3A) (Rakhshandehroo et al 2010). This may suggest that HepG2 cells are less suitable for characterizing the metabolic changes in FAO. Further replicative studies in other hepatic cell lines or animal models are needed to clarify this inconsistency.”

Minor comments

1) Figure 3B. It is incorrect to classify ANGPTL4 as adipocyte differentiation. A better description would be "plasma triglyceride metabolism"

Response:

We apologize for the mistake. In addition to ANGPTL4, our revised manuscript studied many more DEGs that are involved in various biological processes in lipid metabolism (Fig 3A). To reduce the chances of inappropriate or misclassification of the studied genes, we choose to refer to the Reactome database as references and manually assigned several genes if we observed they overlapped in several subsets (Jassal et al 2020). For ANGPTL4, due to lack of other DEGs in triglyceride metabolism identified in our study, we have assigned it to “lipoprotein metabolism” category based on the Reactome pathways R-HSA-174824 and R-HSA-8963899.

2) The paper is quite wordy, in particular the results and discussion. There is no need to elaborate on the basic of PPARs in the discussion (line 413-418). Authors are encouraged to shorten the results and discussion sections.

Response:

We thank the reviewer for this valuable suggestion. Even though we tried to be concise and to the point when drafting, we also found that the discussion sections were unnecessarily wordy. In the revised manuscript, we have totally deleted the two paragraphs (line 413-460 in the initial manuscript) related to PPARA biology. The new discussion section consists one paragraph discussing the possible regulatory mechanisms of the upregulated genes, two paragraphs related to the dysregulation of TFs and their coactivators in lipid metabolism, one paragraph discussing the inconsistencies between our study and previous studies and possible mechanism, and one paragraph on the limitations of the study. For the results section, we also tried our best to shorten the length by using more precise words and not over interpretation of the results and put unnecessary discussion away from the results section. However, because of exploration of the multiple genes in lipid metabolism (Figs 3 and 4), the length of the results section was not significantly shortened. If the length of the manuscript is of problem, we may delete the whole section regarding characterizing the *ELOVL6* promoter (Fig 9) which will not severely change the conclusion of this study.

3) Line 77, Please note that chylomicron remnants carry dietary fatty acids to the liver.

Response:

We thank the reviewer for raising this question. We have changed the description of source of fatty acids as follows “The liver acquires free FAs from three major sources

that are directly dietary intake, lipolysis of TG in adipose tissue and de novo lipogenesis (DNL) in the liver..." in our revision.

4) Line 120, please change principle to principal.

Response:

The incorrect "principle" has been change to "principal".

5) Line 170-173, please rephrase or split in two.

Response:

This sentence has been revised as follows:"CTCF and cohesin usually work as a complex that binds to the same genomic regions to organize higher-order chromatin structures and regulate gene expression (Figure 5F) (Wutz, Várnai et al., 2017). As the expression of both CTCF and cohesin components was not affected by YY1 knockdown, the upregulation of nearby genes may be caused by disruption of the local promoter and enhancer interactions mediated by YY1 and cohesin (Weintraub et al., 2017)."

6) Line 177-179, please rephrase

Response:

This sentence has been revised as follows:" RXRA and HNF4A were observed to frequently bind to the same regulatory regions in ChIP-seq experiments in HepG2 cells (Figure 5H). The downregulation of RXRA and its heterodimeric partner PPARA together with the downregulation of HNF4A and YY1 may have caused the downregulation of nearby genes."

7) Line 703, reference is incomplete.

Response:

The reference has been updated with correct citation information.

References:

Burri L, Thoresen GH, Berge RK (2010) The role of PPAR α activation in liver and muscle. *PPAR Res* 2010 doi:10.1155/2010/542359

Dubois V, Eeckhoutte J, Lefebvre P, Staels B (2017) Distinct but complementary contributions of PPAR isotypes to energy homeostasis. *J Clin Invest* 127(4):1202-1214. doi:10.1172/jci88894

Fernandez-Alvarez A, Alvarez MS, Gonzalez R, Cucarella C, Muntane J, Casado M (2011) Human SREBP1c expression in liver is directly regulated by peroxisome proliferator-activated receptor alpha (PPARalpha). *J Biol Chem* 286(24):21466-21477. doi:10.1074/jbc.M110.209973

Gunn PJ, Green CJ, Pramfalk C, Hodson L (2017) In vitro cellular models of human hepatic fatty acid metabolism: differences between Huh7 and HepG2 cell lines in human and fetal bovine culturing serum. *Physiol Rep* 5(24):e13532. doi:<https://doi.org/10.14814/phy2.13532>

Guzmán C, Benet M, Pisonero-Vaquero S, Moya M, García-Mediavilla MV, Martínez-Chantar ML, González-Gallego J, Castell JV, Sánchez-Campos S, Jover R (2013) The human liver fatty acid binding protein (FABP1) gene is activated by FOXA1 and PPAR α ; and repressed by C/EBP α : Implications in FABP1 down-regulation in nonalcoholic fatty liver disease. *Biochim Biophys Acta, Mol Cell Biol Lipids* 1831(4):803-818. doi:<https://doi.org/10.1016/j.bbalip.2012.12.014>

Houten SM, Violante S, Ventura FV, Wanders RJA (2016) The Biochemistry and Physiology of Mitochondrial Fatty Acid β -Oxidation and Its Genetic Disorders. *Annu Rev Physiol* 78(1):23-44. doi:10.1146/annurev-physiol-021115-105045

Jassal B, Matthews L, Viteri G, Gong C, Lorente P, Fabregat A, Sidiropoulos K, Cook J, Gillespie M, Haw R, et al. (2020) The reactome pathway knowledgebase. *Nucleic Acids Res* 48(D1):D498-d503. doi:10.1093/nar/gkz1031

Lin J, Handschin C, Spiegelman BM (2005) Metabolic control through the PGC-1 family of transcription coactivators. *Cell Metab* 1(6):361-370. doi:<https://doi.org/10.1016/j.cmet.2005.05.004>

Mandard S, Müller M, Kersten S (2004) Peroxisome proliferator-activated receptor α target genes. *Cell Mol Life Sci* 61(4):393-416. doi:10.1007/s00018-003-3216-3

Miller CW, Ntambi JM (1996) Peroxisome proliferators induce mouse liver stearoyl-CoA desaturase 1 gene expression. *Proc Natl Acad Sci U S A* 93(18):9443-9448.

Passilly P, Schohn H, Jannin B, Cherkaoui Malki M, Boscoboinik D, Dauça M, Latruffe N (1999) Phosphorylation of peroxisome proliferator-activated receptor alpha in rat Fao cells and stimulation by ciprofibrate. *Biochem Pharmacol* 58(6):1001-1008. doi:10.1016/s0006-2952(99)00182-3

Pawlak M, Lefebvre P, Staels B (2015) Molecular mechanism of PPAR α action and its impact on lipid metabolism, inflammation and fibrosis in non-alcoholic fatty liver disease. *J Hepatol* 62(3):720-733. doi:<https://doi.org/10.1016/j.jhep.2014.10.039>

Piccinin E, Villani G, Moschetta A (2019) Metabolic aspects in NAFLD, NASH and hepatocellular carcinoma: the role of PGC1 coactivators. *Nat Rev Gastroenterol Hepatol* 16(3):160-174. doi:10.1038/s41575-018-0089-3

Rakhshandehroo M, Hooiveld G, Müller M, Kersten S (2009) Comparative analysis of gene regulation by the transcription factor PPAR α between mouse and human. *PLoS One* 4(8):e6796. doi:10.1371/journal.pone.0006796

Rakhshandehroo M, Knoch B, Muller M, Kersten S (2010) Peroxisome proliferator-activated receptor alpha target genes. *PPAR Res* 2010:612089. doi:10.1155/2010/612089

Yin L, Ma H, Ge X, Edwards PA, Zhang Y (2011) Hepatic hepatocyte nuclear factor 4 α is essential for maintaining triglyceride and cholesterol homeostasis. *Arterioscler Thromb Vasc Biol* 31(2):328-336. doi:10.1161/atvbaha.110.217828

May 20, 2021

RE: Life Science Alliance Manuscript #LSA-2020-00928R

Prof. Claes Wadelius
Science for Life Laboratory, Department of Immunology, Genetics and Pathology, Uppsala University, Uppsala, Sweden
BMC Husargatan 3
Uppsala 75122
Sweden

Dear Dr. Wadelius,

Thank you for submitting your revised manuscript entitled "Multifaceted regulation of hepatic lipid metabolism by YY1". We would be happy to publish your paper in Life Science Alliance pending minor revisions requested by Reviewer 1 and final revisions necessary to meet our formatting guidelines.

Along with the minor revisions requested by Reviewer 1, please also attend to the following to meet the journal's formatting guidelines:

- please upload both your main and supplementary figures as single files
- please add your main, supplementary figure, and table legends to the main manuscript text after the references section
- please add callouts for Figures S1A-B; S3A-H to your main manuscript text
- please include a pbp response with the revision

A. FINAL FILES:

B. MANUSCRIPT ORGANIZATION AND FORMATTING:

Sincerely,

Shachi Bhatt, Ph.D.
Executive Editor
Life Science Alliance
<http://www.lsjournal.org>
Tweet @SciBhatt @LSAJournal

Reviewer #1 (Comments to the Authors (Required)):

During the revision process, the main scope of this submitted paper has been much improved. However, as per this reviewer's opinion, still PPARA story on DNL is not fully convincing. Although authors provided detail image of PPARA regulation on these transcriptions, selected genes like SCD and FADS2 are not only regulated by PPARA but by PPARG, since they share same binding motif-DR1. It is highly likely that PPARG may replace the function of PPARA in luciferase reporter assay of Fig 7/8/9. The possibility of YY1 KD effect on both PPARA and G should be clearly discussed.

Minor Points

1. Did authors actually show YY1 KD regulates lipid accumulation or DNL in HepG2 cells?
2. Please revise lines 72-74. "As the main substrate for TG synthesis...."
3. Please mark "GO" in Fig 2A/2B and "KEGG" in Fig 2C/2D.
4. Please revise line 203, "CHREBP together with its heterodimeric partner MLX and SREBF1". It looks easy to mislead, since SREBF1 is not a heterodimer partner of CHREBP.
5. Please correct truncated references after Fig 10 (page 45 in pdf)

Reviewer #2 (Comments to the Authors (Required)):

The authors did a nice job revising the manuscript. I consider it now suitable for publication.

The response to the reviewers' comments is listed below.

Reviewer #1 (Comments to the Authors (Required)):

During the revision process, the main scope of this submitted paper has been much improved. However, as per this reviewer's opinion, still PPARA story on DNL is not fully convincing. Although authors provided detail image of PPARA regulation on these transcriptions, selected genes like SCD and FADS2 are not only regulated by PPARA but by PPARG, since they share same binding motif-DR1. It is highly likely that PPARG may replace the function of PPARA in luciferase reporter assay of Fig 7/8/9. The possibility of YY1 KD effect on both PPARA and G should be clearly discussed.

We thank the reviewer for these insightful comments. We agree that our current findings can't preclude the possibilities that PPARG may also contributed to the downregulation of the DEGs in DNL pathway (Fig 3A). The downregulation of their common heterodimeric partner RXRA and their common coactivators PGC1A, PGC1B and NCOA1/2/3 for PPARA and PPARG made discerning the relative contribution of PPARA's and PPARG's regulatory effects in YY1-knockdown samples impossible. We have clearly discussed this complexity in the discussion section in line 456-461. It is also high likely that dysregulated YY1 together with elevated expression of PPARG usually observed in the liver of patients with pathological conditions e.g. NAFLD, obesity and diabetes may occupy the same PPARA binding sites identified in Fig 7/8/9 and contributed to the activation of genes in the DNL pathway. Further studies are needed to validate this hypothesis.

Minor Points

1. Did authors actually show YY1 KD regulates lipid accumulation or DNL in HepG2 cells?

This is a very interesting point raised by the reviewer. Previous studies have proposed that either activated DNL or inhibited FAO, which further induce hepatic lipid accumulation due to decreased efficiency of FA utilization, may result in NAFLD progression with dysregulated YY1 (Li et al 2019, Lu et al 2014, Yuan et al 2018). The detailed expression profiling in our study supports the hypothesis that dysregulated metabolic imbalance, especially activated DNL may be the major cause of NAFLD with dysregulated YY1. This is illustrated by the identification of multiple key TFs and their coactivators in DNL, including CHREBP, FXR, PPARA, RXRA, PGC1A, PGC1B, being directly or indirectly regulated by YY1 (Fig 4).

2. Please revise lines 72-74. "As the main substrate for TG synthesis...."

We have amended this sentence as follows: "As the main building blocks of TG, free fatty acids (FAs) play essential roles in the pathogenesis of NAFLD."

3. Please mark "GO" in Fig 2A/2B and "KEGG" in Fig 2C/2D.

We thank the reviewer for pointing this out. We have added appropriate label for each subsection of Figure 2.

4. Please revise line 203, "CHREBP together with its heterodimeric partner MLX and SREBF1". It looks easy to mislead, since SREBF1 is not a heterodimer partner of CHREBP.

We apologize for the possible confusion of this statement. We have corrected the statement as follows: "The expression changes of *SREBF1* together with *CHREBP* and its heterodimeric partner *MLX* were then verified by RT-qPCR which confirmed that only *CHREBP* was significantly downregulated."

5. Please correct truncated references after Fig 10 (page 45 in pdf)

These references were previously cited in Figure legends. They now have been merged with the references in the main manuscript.

Reviewer #2 (Comments to the Authors (Required)):

The authors did a nice job revising the manuscript. I consider it now suitable for publication.

We thank the reviewer for all the insightful and constructive comments and approving our manuscript to be published in *Life Science Alliance*.

References

Li Y, Kasim V, Yan X, Li L, Meliala ITS, Huang C, Li Z, Lei K, Song G, Zheng X, et al. (2019) Yin Yang 1 facilitates hepatocellular carcinoma cell lipid metabolism and tumor progression by inhibiting PGC-1 β -induced fatty acid oxidation. *Theranostics* 9(25):7599-7615. doi:10.7150/thno.34931

Lu Y, Ma Z, Zhang Z, Xiong X, Wang X, Zhang H, Shi G, Xia X, Ning G, Li X (2014) Yin Yang 1 promotes hepatic steatosis through repression of farnesoid X receptor in obese mice. *Gut* 63(1):170-178. doi:10.1136/gutjnl-2012-303150

Yuan X, Chen J, Cheng Q, Zhao Y, Zhang P, Shao X, Bi Y, Shi X, Ding Y, Sun X, et al. (2018) Hepatic expression of Yin Yang 1 (YY1) is associated with the non-alcoholic fatty liver disease (NAFLD) progression in patients undergoing bariatric surgery. *BMC Gastroenterol* 18(1):147. doi:10.1186/s12876-018-0871-2

May 25, 2021

RE: Life Science Alliance Manuscript #LSA-2020-00928RR

Prof. Claes Wadelius
Science for Life Laboratory, Department of Immunology, Genetics and Pathology, Uppsala
University, Uppsala, Sweden
BMC Husargatan 3
Uppsala 75122
Sweden

Dear Dr. Wadelius,

Thank you for submitting your Research Article entitled "Multifaceted regulation of hepatic lipid metabolism by YY1". It is a pleasure to let you know that your manuscript is now accepted for publication in Life Science Alliance. Congratulations on this interesting work.

DISTRIBUTION OF MATERIALS:

Again, congratulations on a very nice paper. I hope you found the review process to be constructive and are pleased with how the manuscript was handled editorially. We look forward to future exciting submissions from your lab.

Sincerely,

Shachi Bhatt, Ph.D.

Executive Editor

Life Science Alliance

<http://www.lsjournal.org>
